# The genomic evolutionary dynamics and global circulation patterns of respiratory syncytial virus

Annefleur C. Langedijk[1,32], Bram Vrancken [2,3,32], Robert Jan Lebbink [4], Deidre Wilkins[5], Elizabeth J. Kelly [5], Eugenio Baraldi[6,7,8], Abiel Homero Mascareñas de Los Santos[9], Daria M. Danilenko [10], Eun Hwa Choi[11], María Angélica Palomino[12], Hsin Chi [13], Christian Keller [14], Robert Cohen[15], Jesse Papenburg[16], Jeffrey Pernica[17], Anne Greenough [7,18], Peter Richmond[19], Federico Martinón-Torres [7,20], Terho Heikkinen[7,21], Renato T. Stein[7,22], Mitsuaki Hosoya[23], Marta C. Nunes [7,24,25], Charl Verwey[24,26], Anouk Evers[4], Leyla Kragten-Tabatabaie[7], Marc A. Suchard [27,28,29], Sergei L. Kosakovsky Pond[30], Chiara Poletto [31], Vittoria Colizza [31], Philippe Lemey [2,33] & Louis J. Bont [1,7,33] ✉ on behalf of the INFORM-RSV Study Group*

Respiratory syncytial virus (RSV) is a leading cause of acute lower respiratory tract infection in young children and the second leading cause of infant death worldwide. While global circulation has been extensively studied for respiratory viruses such as seasonal influenza, and more recently also in great detail for SARS-CoV-2, a lack of global multi-annual sampling of complete RSV genomes limits our understanding of RSV molecular epidemiology. Here, we capitalise on the genomic surveillance by the INFORM-RSV study and apply phylodynamic approaches to uncover how selection and neutral epidemiological processes shape RSV diversity. Using complete viral genome sequences, we show similar patterns of site-specific diversifying selection among RSVA and RSVB and recover the imprint of non-neutral epidemic processes on their genealogies. Using a phylogeographic approach, we provide evidence for air travel governing the global patterns of RSVA and RSVB spread, which results in a considerable degree of phylogenetic mixing across countries. Our findings highlight the potential of systematic global RSV genomic surveillance for transforming our understanding of global RSV spread.

With the recent approval of the first-ever respiratory syncytial virus (RSV) vaccines and the monoclonal antibody (mAb) nirsevimab for the prevention of RSV in all infants[1], our understanding of the global transmission dynamics of RSV becomes increasingly important. An important unsolved question is to what extent RSV epidemics are fuelled by local persistence from a previous epidemic versus that of

viral seeding from other geographic areas. A better understanding of the global circulation dynamics and local persistence is crucial for RSV surveillance and prevention.

Viral genetic sequence data may offer valuable information to aid in testing predictors of spread and to empirically develop and validate epidemiological models. A challenge for reconstructing viral spread

A full list of affiliations appears at the end of the paper. *A list of authors and their affiliations appears at the end of the paper. ✉e-mail: l.bont@umcutrecht.nl

through space and time from genetic data has been the lack of a systematic and comprehensive global sampling of whole genomes from circulating RSV lineages. Current such sampling efforts include the global multiyear multicentre INFORM-RSV study and the Global RSV Surveillance Programme of the World Health Organisation (WHO). The INFORM-RSV study combines large-scale full genome sequencing and a global coverage over multiple RSV seasons to provide a molecular reference of RSV strains and sequence variability[2]. The best way of mapping genomic evolutionary dynamics of RSV is by analysing nucleotide substitutions of the complete genome. Previously selective pressure analyses with samples from the 2001–2011 time period showed that RSV genes consist predominantly of negatively selected and neutrally evolving sites. Only the G gene encoding for the surface glycoprotein G stood out in terms of detectable positive selection[3]. The primary role of the G protein is to attach virions to cell surfaces through interaction with host cell attachment factors[1,4]. The genetic factors that impact the replacement dynamics remain poorly understood and a full-genome perspective on the adaptive evolution of RSV is needed to reveal which other genomic variations affect the fitness of strains.

While sequencing efforts have been implemented on a large scale for SARS-CoV-2, systematic sequencing of RSV is still at an early, small scale stage. For respiratory viruses such as seasonal SARS-CoV-2 and influenza, human air-based travel (flight) has been shown to be an important driver of global circulation[4–8]. Air travel may also shape seasonal RSV dynamics. RSV molecular epidemiology data from Kenya showed that several new variants are introduced every epidemic season[9–15]. The interspersed nature of sequences from Kilifi and other parts of Kenya indicates a degree of mixing of lineages, which in turn suggests that air travel may be an important driver of spread. However, the global circulation patterns of RSV have remained unexplored. Therefore, we integrated human movement patterns with whole genome sequences from RSV samples that were collected in 17 countries worldwide over three RSV seasons (2017–2020) prior to the COVID-19 pandemic. Here, we show that air travel predicts global RSV spread. Travel restrictions due to COVID-19 have not affected the current analysis.

## Results
### Circulating genotypes
We obtained 1282 complete RSV genome sequences collected over a period of three years from 17 countries worldwide enroled in the INFORM-RSV study. We complemented these sequences with 1180

publicly available sequences from NCBI GenBank sampled within the same time interval. All RSVA and RSVB genomes in the genotyping datasets cluster among strains that were typed as A23 and B6. For this reason, the genotyping alignments were appended with strains of genotype A22 (RSVA) and B5 (RSVB) that served as outgroups for rooting the maximum likelihood (ML) trees. Applying previously established genotyping criteria show that genotypes A23 and B6, from which the currently circulating strains have evolved, can be reclassified into a set of 25 RSVA and 2 RSVB genotypes (Fig. 1). Variants with a duplication in the G gene have emerged[16]. These variants appear to have a fitness advantage[17] and have started to replace previously circulating strains. This observation is reflected in our data, as 100% of the sequenced RSVA and RSVB isolates carry these duplications.

### Comparable site-specific diversifying selection in RSVA and RSVB
To identify positively selected sites in the coding genes of the RSV genome, we employ three different methods (FUBAR, MEME, and RC, cfr. Methods) that aim to capture different aspects of site-specific selection and report sites that were identified by at least two of these methods. Using this approach, we identify 28 positively selected amino acid sites in RSVA. Of these, 21 are located in the G protein, one in the F protein, and six in the L protein. Eight of the G protein sites and one L protein site are supported by all three methods. We obtain a similar number ($n = 26$) and distribution of positively selected sites in RSVB, with 18 sites in the G protein, two in the F protein, and six in the L protein. Eight of the G protein sites and one F protein site are supported by all three methods. Three of the positively selected sites are identified at the exact same amino acid position in the G protein of RSVA and RSVB (amino acid positions 136, 274, 310). However, the amino acid position on the linear protein sequence for RSVA may not necessarily be the same as for RSVB in the protein crystal structure. Substitutions in positions under positive selection are found on different branches of phylogeny, which is consistent with the expectation under diversifying selection (Figure S1 and S2).

### Both RSVA and RSVB genealogies are shaped by non-neutral population turnover
RSV evolution may be shaped by selection for variants with higher replicative fitness and variants that evade host immune responses[18]. The latter is indicated by the site-specific selection analyses that

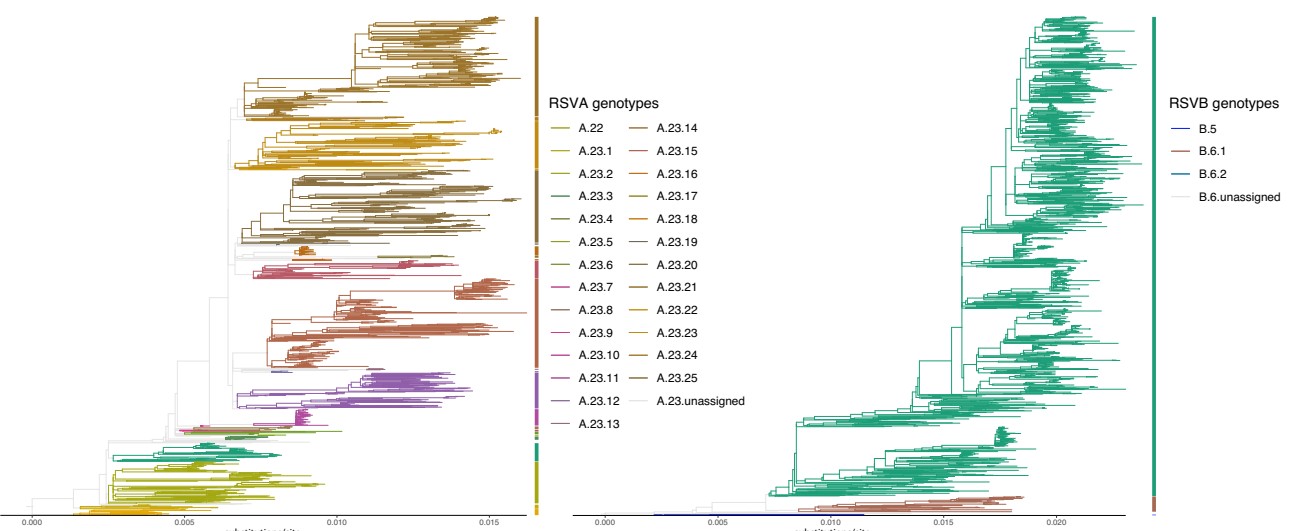

**Fig. 1 | Maximum likelihood reconstructions of RSVA (1482 genomes/taxa; 2006–2020) and RSVB (1543 genomes/taxa; 1997–2020) complete genome phylogenies and genotypes identification.** Lineages that are not assigned to a genotype are shown in light grey. The SH-aLRT and UFB support values for the genotypes are provided in Supplementary Table S1.

identify the G gene as the major target of diversifying selection[3,18]. However, earlier testing has found that only RSVB tree shapes inferred from complete genome data deviate from what we expect under neutrality[3,18]. Now that considerably more complete genome data are available, we revisit the genealogical testing using posterior predictive simulation[19]. We employ the genealogical Fu and Li statistics as well as a trunk length proportion statistic as tree shape statistics (see Methods). We plot bivariate distributions for these statistics based on the genealogies inferred from the genomic data and the equivalent genealogies simulated under neutrality accommodating for potentially complex histories of population size change (Fig.S3). Both RSVA and RSVB show significant deviations from neutrality, with a more pronounced deviation for RSVB as compared to RSVA.

## Global RSV circulation patterns are shaped by human air travel

To explore the factors that shape RSV global circulation, we apply a Bayesian phylogeographic approach that models the movement of virus lineages between a set of discrete locations[20]. This process is generally parameterised in terms of transition rates for all pairs of locations. Here, we use an extension of the discrete phylogeographic model that parameterises these transition rates as a function of a number of potential predictors[5]. This generalised linear model (GLM) parameterisation allows estimating the contribution of each predictor to the spatial diffusion as a coefficient (on a log scale). In addition, the model includes boolean indicator variables that determine the in- or exclusion of predictors allowing to estimate their inclusion probability. Here, we report the posterior distribution of the product of the log coefficient and inclusion probability for each predictor; positive estimates indicate a positive association between predictors and diffusion intensity while the opposite is true for negative estimates. As predictors, we consider human air travel, population size, geographic distances, and latitude differences (see Methods). Our analyses consistently support human air travel as a strong predictor of RSV global spread at both the country (strongly positive estimates, Fig. 2) and continental level (Fig. S4) for RSVA and RSVB separately, as well as for a model applied to both RSVA and RSVB data sets combined. The support for air travel is robust to the inclusion of sample sizes as predictors. Other candidate predictors occasionally find support, but not consistently so, suggesting that these other predictors could for example be attributed to sampling variability. For instance, the human population size at the origin location is estimated to have a negative log coefficient for its effect size in the RSVB analyses. This may be explained by the fact that the most populous countries, such as China and India, are represented by only a few genomes that are distributed as singletons in the phylogeny, thereby resulting in an underestimation of their potential role as origin locations in the global circulation dynamics. In fact, these two locations specifically have been shown to be important for persistence and global dissemination of seasonal influenza viruses[4]. Therefore, better global coverage will be needed to characterise the role of undersampled countries in RSV circulation and how they may relate to demographic characteristics.

While the phylogeographic data sets include genomes sampled between 2012 and 2020, the INFORM-RSV study contributes to the most recent years (2017–2020) of sampling. To determine how these data contribute to predictor support, we also apply a time-inhomogeneous GLM-diffusion model distinguishing between the five most recent years and the 5-year time period before that (Fig. S5). This illustrates that the support for air travel is consistently found for the recent time period whereas this is less convincing or less consistent across analyses in the earlier time period. This demonstrates how systematic global sampling contributes to the opportunity to identify meaningful patterns of RSV spatial spread.

## Phylogeographic reconstructions indicate extensive geographic mixing

RSV spread by air travel offers the opportunity for substantial geographic mixing of viral lineages between locations. To assess geographic mixing, we use recently proposed entropy-based phylogeographic summaries for the genome sampling in the most recent pre-pandemic INFORM-RSV season (2019–2020). Specifically, we summarise normalised entropy measures or the phylogeographic clustering by country, reflecting the degree of phylogenetic interspersion of country-specific lineages (Fig. 3), and the number of unique lineages associated with each country circulating at the start of the most recent RSV season (see Supplementary Files S1 and S2 for the MCC summary trees from the evolutionary reconstructions underlying these inferences). Some countries have different results for RSVA versus RSVB, which could be explained by the fact that whether a lineage grows to be a persistent one is a stochastic event even if particular countries would be more prone to persistent circulation. This normalised entropy ranges between 0, reflecting no intermixing of viruses from different countries, and 1, reflecting a clustering that is randomised with respect to country of sampling.

For RSVA, we infer relatively high entropy estimates, with 13 out of 15 estimates above 0.8. For the Netherlands for example, we estimate entropy of 0.88 [95% highest posterior density interval (HPD) 0.82,0.94] and 10 [95% HPD 8,12] unique lineages circulating at the start of the most recent season (2019–2020), which together are represented by 23 sampled genomes in the final season. With an entropy estimate of 0.33 [95% HPD 0.28,0.38], South Africa appears to be an exception to the pattern of relatively extensive mixing. While we estimate a substantial number of unique South African lineages at the start of the final season (26 [95% HPD 21,30]), there is also a substantial degree of clustering of the 58 genomes sampled from that season, with 50 out of 58 samples belonging to a large South African cluster including also samples from the previous season (Fig.S6). Similarly high entropies are estimated for RSVB in most countries. While two more mean estimates fall below 0.8, their credible intervals are broad. Although the mean entropy estimate for South Africa is also <0.8 for RSVB, the deviation from countries with high entropy values is far more limited. Overall, these estimates suggest a substantial global geographic mixing of both RSVA and RSVB.

## Discussion

Optimised surveillance and prevention of RSV infection at a global scale relies on our understanding of its spread. Here, we combine existing RSV genomic data and new full genomes from a systematic global sampling effort with empirical data on human mobility, demography and a proxy for synchronicity of RSV seasonality to evaluate which factors shape global RSV circulation. We show that air travel predicts global RSV spread, similar to what has been demonstrated for influenza H3N2[5,8], influenza H1N1[4], and recently SARS-CoV-2[6]. Additional sampling efforts (including those within the framework of the ongoing INFORM-RSV study) are expected to generate more densely sampled genomic data. This will increase the resolution of phylogeographic reconstructions and it will likely allow testing predictors at other spatial scales where other forms of mobility could also shape RSV circulation. Understanding RSV spread is also important in the light of monitoring for escape mutations to emerging prophylactic approaches to RSV, as our findings show these have the potential to spread rapidly on a global scale.

Human air travel increases the likelihood of infectious diseases spreading rapidly between countries and continents[21]. We speculate that air traffic could be a mechanism of RSV transmission. It is still unclear how patients acquire viral respiratory disease in the context of air travel, and the prevalence of RSV in airplane passengers has not been studied. Previous research showed that almost one-half of all patients with clinical symptoms upon travel turn were infected with

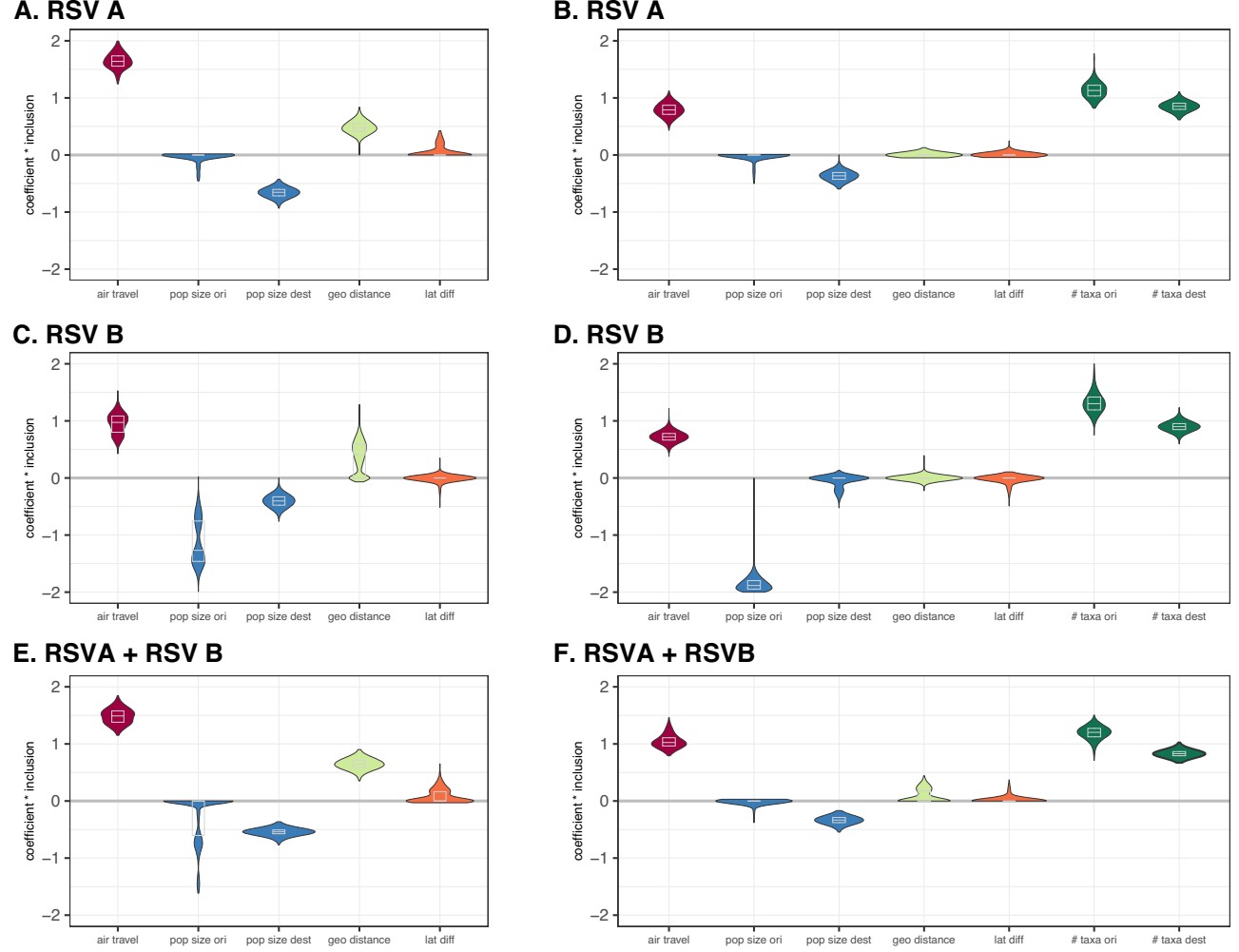

**Fig. 2 | Posterior estimates of time-homogeneous predictor contributions to RSV diffusion between countries.** The predictors include the number of passengers travelling by air between each pair of countries represented in the data set (air travel, in dark red), population size at the origin and destination location (pop size ori & pop size dest, in blue), geographic distance (geo distance, in light green), absolute differences in latitude (lat diff, in dark orange) and sample sizes at the origin and destination locations (# taxa ori & # taxa dest, in dark green). The Y-axis represents the product of the coefficient (on a log scale) and the inclusion probability for the predictors (coefficient * Inclusion). (**A**, **B**: RSVA. **C**, **D**: RSVB. The plots on the left and right distinguish between analyses without and with sample size predictors respectively. **E** and **F** summarise the estimates for a single GLM-diffusion model applied to the combined RSVA and RSVB data sets at the country level. The grey boxes in the violin plots represent the median and quantile estimates. Violin plots are based on n = 507 (**A**), n = 535 (**B**), n = 45002 (**C**), n = 45002 (**D**), n = 452 (**E**) and n = 452 (**F**) post-burnin samples from the respective MCMC chains. Source data are provided as a Source Data file.

respiratory viruses[22,23]. Other evidence suggests that SARS-CoV-2 is transmitted during air travel[24,25]. Global concerns such as the emergence of Ebola Virus Disease in West Africa[26] and novel SARS-CoV-2 variants[27] have already led to a number of protocols implemented at airports of departure or arrival (e.g. testing, genomic surveillance, quarantines, etc.). As global connectivity has increased, so has the potential for RSV to spread across countries. Before the COVID-19 pandemic, over four billion passengers travelled by airplane annually and this number is likely to double by 2036. We expect the main mechanism of global spread to be spread at the country of arrival, mostly due to travellers infected in the community and bringing the infection from a seeding area where the epidemic is ongoing to the destination country. We show that seasonal RSV epidemics are likely fuelled by many independent introductions. However, the exact source locations cannot be identified with our data.

Our reconstructions provide some evidence of local RSV persistence in South Africa. These data build on earlier evidence of clustering and strong selective pressure for both RSVA and RSVB in South Africa[28]. RSV clustering in South Africa resembles data on influenza A which persisted in West Africa for almost two years[29]. Extensive spatial

mixing of influenza A by air travel was observed in West Africa, perhaps because of its relatively lower connection within the global air transportation network. The climatic variability may also have contributed to the influenza persistence generating temporal overlap among epidemics[29].

Currently, several genotype definitions are used in parallel and there is no universal approach to classify virus genetic diversity[30]. Therefore, genotyping based on complete genome sequences, instead of genotyping based on nucleotide sequence variability of subgenomic regions (mostly the G gene), can improve the RSV surveillance field by providing a more coherent classification. By focusing on active virus lineages and those spreading to new locations, this universal nomenclature would assist in tracking and understanding the patterns and determinants of the global spread of RSV. For SARS-CoV-2, a similarly proposed nomenclature represents an important asset to the field[31]. We hope that our study will motivate large-scale implementation of whole genome sequencing for RSV surveillance.

Site-specific selection analyses identified the G gene as the main target of diversifying selection. When compared to influenza with its ladder-like phylogeny and strong turnover, positive selection for RSV

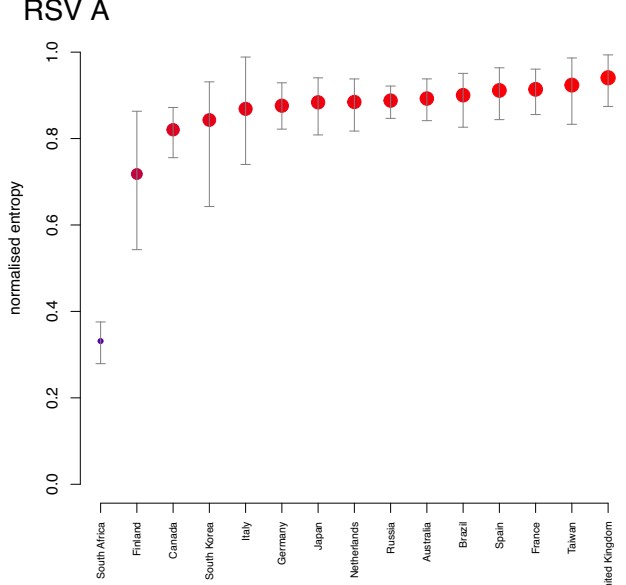

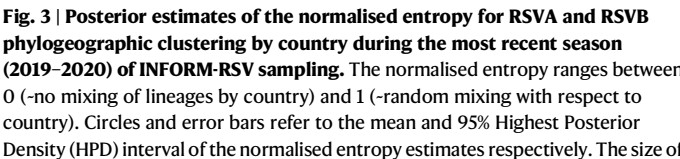

**Fig. 3 | Posterior estimates of the normalised entropy for RSVA and RSVB phylogeographic clustering by country during the most recent season (2019–2020) of INFORM-RSV sampling.** The normalised entropy ranges between 0 (-no mixing of lineages by country) and 1 (-random mixing with respect to country). Circles and error bars refer to the mean and 95% Highest Posterior Density (HPD) interval of the normalised entropy estimates respectively. The size of the circles is proportional to what fraction of the highest mean estimate each average estimate represents. The same is indicated by the colours of the circles, which range from blue for an average estimate that represents 0% of the highest value to bright red for the highest mean estimate. Entropy estimates are based on n = 901 post-burnin samples from the stationary MCMC chain. Source data are provided as a Source Data file.

is less strong. Our results confirmed that the RSV genome is largely conserved, with the exception of the highly variable G gene. We have identified different positions under selective pressure for RSV A and B reporting on positive selection on the L gene at amino acid position 146, 624, 1725, 1748, 2111, and 2113 for RSVA and 560, 1712, 1718, 1719, 1759 and 2019 for RSVB, which may represent epitopes under pressure of adaptive immunity[32]. Immunological studies are required to confirm adaptive immune responses are developed during RSV infections against these epitopes on the L gene.

Strengths of this study are the sample size, the use of complete genomes, and a broad geographic coverage over a period of many years. Another strength is that our study only included prepandemic RSV sequences and mobility data, as COVID-19 drastically impacted human air travel. An important limitation of our study is lack of data from most of the African continent, as well as from specific large countries including China and India. Additionally, the sample size within countries was too small to explain short-distance spread of RSV. Broader and denser coverage is likely to reveal additional predictors at different scales of transmission.

RSV research and therapeutics are rapidly advancing with the recent approval of nirsevimab and two vaccine for older adults, which might be shortly followed by the approval of a maternal vaccine[1]. Surveillance of RSV may be particularly important in the wake of these vaccines, given the potential for increased immunologic pressure on RSV F. The integration of epidemiological and phylogenetic approaches has received great attention for other viruses because of its potential to uncover mechanisms of pathogen emergence, evolution, and spread. By capturing the spatial spread of RSV, our reconstructions of spatial evolutionary history shed light on viral persistence and transmission dynamics. We demonstrate that the use of human air travel data together with viral genetic data provides a powerful model to describe global spread of RSV. This work also provides a baseline of RSVA and RSVB genome evolution before the widespread use of immunisation programmes, and the new genome data will constitute a key resource for further extensive research in the field of RSV epidemiology.

## Methods

### Clinical samples
The INFORM-RSV study is a prospective, multiyear, multicentre, global clinical study enroling children with medically-attended RSV infection under the age of 5 years. Details about the study design and protocol have been previously described[2]. In summary, RSV positive nasal samples were collected from November 2017 to March 2020 at 18 hospitals in 17 countries globally. Whole genome sequencing was performed at the UMC Utrecht using the Illumina NextSeq 500 platform (details have been published a separate methodology paper[2]) and annotated with sampling data and country. Whole genome sequences derived from the first three seasons of the INFORM-RSV study are available at GenBank.

### Ethical approval and consent
We declare that the planning, conduct, and reporting from this study was in line with the Declaration of Helsinki, as revised in 2013. Informed consent was obtained from parent(s) or legal representative(s) prior to sample collection in accordance with the International Conference on Harmonization Guideline on Good Clinical Practice E6 (ICH-GCP) and applicable national and international regulatory requirements. The INFORM-RSV study has been approved by the ethics committees of all 18 participating sites: The Netherlands: The Medical Research Ethics Committee of the UMC Utrecht (reference number WAG/mb/17/016170); Italy: Ethics Committee for Clinical Testing of the Province of Padova of the Padova Hospital (no. 345 of 27/10/2016); Russia: The Department for Science, Innovation Development and Management of Health and Biological Risks, Ministry of Health of the Russian Federation; Germany: Ethics Committee of the Medical Faculty of the Philipps University Marburg; France: Ethics Committee Southwest and Overseas of the Créteil Intercommunal Hospital Centre (ID-RCB No.: 2018-A02360−55 (file 1− 18−73); Spain: Ethics Committee for Research Santiago-Lugo of the Hospital Centre University of Santiago (registration code 2017/397); South Korea: Medical Research Committee of the Seoul National University Hospital; Finland: Ethics Committee of the Hospital District of Southwest Finland, Turku;

Australia: Human Research Ethics Committee of the Perth Children's Hospital; Brazil: The Research Ethics Committee of the Centro INFANT at Pontificia Universidade Catolica de Rio Grande do Sul (opinion number 2,569,872); Canada: Hamilton Integrated Research Ethics Board of the McMaster University; Canada: Research Ethics Board of the McGill University Health Centre; South Africa: Human Research Ethics Committee of the University of the Witwatersrand Johannesburg (no. M170966); Japan: Research Ethics Committee of the Fukushima Medical University (no. 29212); The United Kingdom: Health Research Authority of the King's College Hospital (no. 17/EM/0469); Taiwan: Mackay Memorial Hospital Institutional Review Board (no. 19MMHIS171e); Chile: Ethics Committee for Research on Human Subjects of the Faculty of Medicine, University of Chile; Mexico: Ethics Committee of the University Autónoma De Nuevo León, Faculty of Medicine.

### Data set compilation

Sequence data on the F protein of RSVA and RSVB from the INFORM-RSV study have previously been published[33]. However, the current data represent the first whole genome sequences which were complemented with a selection of publicly available RSV sequences downloaded from NCBI GenBank on April 21st 2021. These were first size-selected (only those of length without N >= 10k bases were kept for further analyses, n = 2865/27417 or 10.4%) and typed as RSVA or RSVB. After alignment with MAFFT v.7.475[34] and manual verification using AliView v.1.26[35], RDP5[36] was used to clean the RSV A and RSVB alignments from putative recombinant sequences. Next, only sequences with known country of sampling and sampling date known up to the year or more precise were retained for further analyses. The resulting alignment served to obtain a maximum likelihood tree with branch support estimated with the SH-aLRT test[37] as implemented in IQtree v.2.1.2[38]. From this tree, a well-supported subtree containing all INFORM-RSV sequences was selected for downstream analyses (Figs. S7 and S8).

### Circulating genotypes

We investigated whether the additional genomic diversity from the INFORM-RSV samples warrants a reclassification. For this we adhered to the RSV type-specific patristic distance thresholds suggested by ref. 30 but assess clade support with the computationally more efficient SH-aLRT and UFB branch support tests, and require minimal support values of 80 (SH-aLRT) and 90 (UFB). The criteria for genotype delineation put forward by ref. 30 involve a patristic distance and a clade support threshold. This definition implies that genotypes form monophyletic clades in which a limited number of genetic differences has accrued. It can therefore be anticipated that, as evolution continues, a clade that was formerly classified as a single genotype can diversify into a set of new genotypes.

TempEst v.1.5.3[39] was used to identify sequences that represented outliers in a regression of root-to-tip divergence as a function of sampling time. To this end, an operational definition of outliers was used: outliers were defined as sequences for which the residual of the regression of root-to-tip genetic distance against sampling time falls outside the 99% credible interval of residuals, which was derived using the CODA R package[40,41]. 13 outliers were removed from the RSVA and RSVB data sets. This increased the correlation between the root-to-tip distance and sampling time from 0.94 to 0.95 for RSVA and from 0.79 to 0.83 for RSVB. Likewise, the $R^2$ of the regression increased from 0.89 to 0.91 for RSVA and from 0.63 to 0.70 for RSVB. The resulting data sets, with 1213 taxa for RSVA and 1223 taxa for RSVB, were used for phylogeographic reconstruction and genotype classification[30]. For the latter, a maximum likelihood tree was estimated using IQtree[38] with ModelFinder[42] and branch support was evaluated with the SH-aLRT and ultra-fast bootstrapping (UFB) procedures. Genotypes were

called using an in-house developed R[41] script that capitalises on several packages (treeio, phytools, geiger).

A down-sampled data set was created for site-specific selective pressure analyses. For this, within-country transmission networks were downsized to a randomly chosen taxon according to a two-step procedure. First, within-country transmission networks were identified as clades with perfect SH-aLRT support for which all taxa were from the same country[43] based on a midpoint rooted maximum likelihood tree (obtained with IQtree v.2.1.2[38]) from the phylogeo-datasets. Next, this reduced data set was used for estimating time-calibrated evolutionary histories with the Bayesian Evolutionary Analysis by Sampling Trees software (BEAST v1.10)[44] along with the high-performance BEAGLE v.3.2.0 library for computational efficiency[45]. The RSVA and RSVB data sets were equipped with the same evolutionary models. To capture the nucleotide substitution process while allowing for differences between the coding and non-coding genome regions, a General Time Reversible (GTR) model with Γ-distributed among site rate variation[46,47] was specified for either region. The estimated rate of evolution was informed by the amount of evolution that accrued over the sampling time differences, and the rate was allowed to vary among lineages through a relaxed clock model with lognormally distributed branch rates[48]. The demographic history was modelled with the flexible skygrid tree prior[49] with changes in the relative genetic diversity over time allowed at 6-month intervals between January 1st 2020 and January 1st 2005. Within country transmission chains were now identified as clades of taxa from the same country with perfect posterior support.

### Phylogeographic inference

Time-calibrated evolutionary histories were estimated from the phylogeography data sets using the Bayesian Evolutionary Analysis by Sampling Trees software (BEAST v1.10)[44] along with the high-performance BEAGLE v.3.2.0 library for computational efficiency[45]. The same models as for identifying within-country transmission networks (see above) were specified. Mixing and convergence properties of the Markov Chain Monte Carlo simulation were inspected using Tracer v1.7[50]. Maximum Clade Credibility (MCC) summary trees were obtained with TreeAnnotator (distributed with BEAST v.1.10) and visualised in FigTree v.1.4[51]. Continuous parameter estimates are summarised as means and 95% highest posterior density intervals (95% HPDs).

### Generalised linear mixed model

To test for predictors of the global spatial diffusion process, we applied a generalised linear model (GLM) parameterisation of the discrete phylogeographic model[5]. Briefly, this model parameterises the log transition rates between pairs of locations as a function of potential predictors. Each predictor is associated with an estimable log effect size and inclusion probability. We reported the posterior estimates for the product of these parameters for our analyses. We applied this model both at the country and the continental level and employ a set of 1000 time-scaled trees sampled evenly throughout the post-burning posterior as empirical tree distributions for both RSVA and RSVB.

For the reconstruction at continental level, taxa were assigned to Africa, Asia, Europe, North America, Oceania or South America based on the WHO region classification. Specifically, taxa from the Sub-Saharan Africa and Northern Africa regions were categorised as African. Taxa from the Western, Central, Southern Eastern and South-Eastern Asia regions were categorised as Asian. Taxa from the Caribbean, Central and Northern America regions were categorised as North American. South American countries were categorised as South American. Countries from Melanesia, Micronesia, Polynesia together with Australia and New Zealand were categorised as Oceania. Taxa from Eastern, Western, Northern and Southern Europe were binned as European.

As predictors, we included passenger fluxes (i.e. the number of passengers travelling by air between countries and continents provided by the International Air Transport Association (IATA)[52] for the period 2019–2020), population size (for 2019)[53] at the origin and destination location, geographic distance and absolute difference in latitude (as proxy for synchronicity in northern or southern hemisphere transmission). For the geographic distances and absolute latitude differences, latitude and longitude coordinates representing the countries' midpoints were downloaded from the Dataset Publishing Language as provided by Google[50]. Geographic distances were calculated using the Haversine formula. At the continental level, we used data for the countries from which genome samples are included in the analyses. In additional analyses, we assessed the sensitivity of predictor support with respect to sampling heterogeneity by also including sample size at the origin and destination location as potential predictors. Analyses were performed for both RSVA and RSVB separately, but we also ran the inference applying a single GLM-diffusion model to both data sets to examine the shared signal in both. Finally, for the country-level analyses we also applied a time-inhomogeneous version of the model[54] partitioning the evolutionary history in an epoch before and after 5 years since the most recent sampling time. These analyses were performed to examine which time period was informing the predictor support.

## Posterior summaries of geographic mixing

To quantify the degree by which RSV clustering is structured by country, we used a normalised entropy measure recently proposed by ref. 6. We focused on the most recent season (2019–2020) because the phylogenetic clustering of these samples and their degree of phylogenetic interspersion is expected to be maximally informed by the INFORM-RSV sampling during the two previous seasons. For each country, we considered a time interval that encompasses the sampling from that recent season and goes back to the end of the previous season for that country. The start and end months of RSV seasons were determined by the relative infection intensities per month for each country. In these time intervals, we summarised the times associated with contiguous partitions of a tree estimated to be in each country. Based on these time estimates we computed a normalised Shannon entropy for each country:

$$-\frac{1}{\ln(n)}\sum_{i}^{n} p_i \ln(p_i)$$

Where $p_i$ is the proportion of time associated with that country for partition $i$ of the tree, and $n$ represents the number of partitions for that country in the tree. In case all genomes sampled during the most recent season in a specific country would form a single cluster (partition) in the phylogeographic tree, the entropy measure is expected to be $\approx 0$. When none of the genomes from the same country would cluster together, and hence are interspersed with genomes from other countries, the measure is expected to be $\approx 1$. We used this measure to summarise the posterior distribution of phylogeographic reconstructions for the analysis with a single time-inhomogeneous GLM-diffusion model shared by both RSVA and RSVB (without sample size predictor). To aid interpretation of the entropy measures, we also summarised the number of unique lineages circulating in each country at the start of the most recent season. Multiple branches associated with the same country sharing a common ancestor with that country state after the end of the previous season are considered to constitute a single unique lineage[6]. We also attempted to summarise whether these unique lineages represented new introductions or persisting lineages since the end of the previous season for each country[6], but this results in uninformative estimates because of an insufficiently dense sampling each season and lack of global coverage. Specifically, lineages from the last season often coalesced with other lineages earlier than the previous season, biasing the estimates towards persistence.

## Identification of positively selected sites

Following recommendations by Kosakovsly Pond and Frost[29], we identified positively selected sites using different complementary approaches. Specifically, we employed the fast unconstrained Bayesian approximation (FUBAR) and the mixed effects model of evolution (MEME) approach implemented in HyPhy and the renaissance counting (RC)[55] approach implemented in BEAST. For FUBAR, we used the variational Bayes approximation and the default threshold of a posterior probability >0.9 for sites to be identified as subject to diversifying positive selection. For MEME, we used the default $p$-value threshold of 0.1 for testing for selection and we restrict the test to internal branches. For RC, we specified a skygrid coalescent prior, an uncorrelated relaxed clock model, and a GTR model for each codon position. We considered sites to be positively selected if the site-specific empirical Bayes estimate of the nonsynonymous to synonymous rate ratio (dN/dS) results in a lower 95% HPD interval boundary that is larger than 1 and if the mean dN/dS estimate is larger than 1.5. We only reported sites as positively selected if they are identified by at least two of the three approaches used.

**Genealogical neutrality tests.** To evaluate whether RSV evolution adheres to neutral evolution, we employed a model-based Bayesian procedure that distinguishes between the effects of demography from the effects of selection[19]. Specifically, we employed the posterior distribution from the genealogical inference produced by BEAST and perform posterior predictive simulation of genealogies under neutral coalescent models accounting for potentially complex demographic histories. For the latter, we adopt the skygrid coalescent model. For posterior predictive simulation under this model, we fit skew normal distributions to the estimates of the interval-specific population sizes and use these in an MCMC simulation procedure. By comparing the genealogical shapes of the inferred tree distribution to that obtained by the posterior predictive simulation using summary statistics, we tested for significant departures from neutral evolution. Here we used two genealogical summary statistics: i) the genealogical Fu and Li statistic (DF), which compares the length of terminal branches to the total length of the coalescent genealogy[19], and ii) the ratio of the trunk (or backbone) length over the entire tree length. The concept of a trunk, representing the lineage(s) that persist(s) through time, has frequently been used in characterisation of the viral population turnover dynamics[56,57], with viruses like human seasonal influenza that experience strong selective pressure to escape antibody responses showing pronounced trunk and short-lived side branches.

## Reporting summary

Further information on research design is available in the Nature Portfolio Reporting Summary linked to this article.

# Data availability

The whole genome sequencing data generated in this study have been deposited in GenBank under the accession codes PPP376262 to PP377590 listed below. Alignments, predictor data, Source Data and BEAST XML files used for this work are publicly available on GitHub (https://github.com/bramvrancken/RSV_INFORM.git, https://doi.org/10.5281/zenodo.8422698). PP376262, PP376263, PP376264, PP376265, PP376266, PP376267, PP376268, PP376269, PP376270, PP376271, PP376272, PP376273, PP376274, PP376275, PP376276, PP376277, PP376278, PP376279, PP376280, PP376281, PP376282, PP376283, PP376284, PP376285, PP376286, PP376287, PP376288, PP376289, PP376290, PP376291, PP376292, PP376293, PP376294, PP376295, PP376296, PP376297, PP376298, PP376299, PP376300, PP376301, PP376302, PP376303, PP376304, PP376305, PP376306,

PP376307, PP376308, PP376309, PP376310, PP376311, PP376312, PP376313, PP376314, PP376315, PP376316, PP376317, PP376318, PP376319, PP376320, PP376321, PP376322, PP376323, PP376324, PP376325, PP376326, PP376327, PP376328, PP376329, PP376330, PP376331, PP376332, PP376333, PP376334, PP376335, PP376336, PP376337, PP376338, PP376339, PP376340, PP376341, PP376342, PP376343, PP376344, PP376345, PP376346, PP376347, PP376348, PP376349, PP376350, PP376351, PP376352, PP376353, PP376354, PP376355, PP376356, PP376357, PP376358, PP376359, PP376360, PP376361, PP376362, PP376363, PP376364, PP376365, PP376366, PP376367, PP376368, PP376369, PP376370, PP376371, PP376372, PP376373, PP376374, PP376375, PP376376, PP376377, PP376378, PP376379, PP376380, PP376381, PP376382, PP376383, PP376384, PP376385, PP376386, PP376387, PP376388, PP376389, PP376390, PP376391, PP376392, PP376393, PP376394, PP376395, PP376396, PP376397, PP376398, PP376399, PP376400, PP376401, PP376402, PP376403, PP376404, PP376405, PP376406, PP376407, PP376408, PP376409, PP376410, PP376411, PP376412, PP376413, PP376414, PP376415, PP376416, PP376417, PP376418, PP376419, PP376420, PP376421, PP376422, PP376423, PP376424, PP376425, PP376426, PP376427, PP376428, PP376429, PP376430, PP376431, PP376432, PP376433, PP376434, PP376435, PP376436, PP376437, PP376438, PP376439, PP376440, PP376441, PP376442, PP376443, PP376444, PP376445, PP376446, PP376447, PP376448, PP376449, PP376450, PP376451, PP376452, PP376453, PP376454, PP376455, PP376456, PP376457, PP376458, PP376459, PP376460, PP376461, PP376462, PP376463, PP376464, PP376465, PP376466, PP376467, PP376468, PP376469, PP376470, PP376471, PP376472, PP376473, PP376474, PP376475, PP376476, PP376477, PP376478, PP376479, PP376480, PP376481, PP376482, PP376483, PP376484, PP376485, PP376486, PP376487, PP376488, PP376489, PP376490, PP376491, PP376492, PP376493, PP376494, PP376495, PP376496, PP376497, PP376498, PP376499, PP376500, PP376501, PP376502, PP376503, PP376504, PP376505, PP376506, PP376507, PP376508, PP376509, PP376510, PP376511, PP376512, PP376513, PP376514, PP376515, PP376516, PP376517, PP376518, PP376519, PP376520, PP376521, PP376522, PP376523, PP376524, PP376525, PP376526, PP376527, PP376528, PP376529, PP376530, PP376531, PP376532, PP376533, PP376534, PP376535, PP376536, PP376537, PP376538, PP376539, PP376540, PP376541, PP376542, PP376543, PP376544, PP376545, PP376546, PP376547, PP376548, PP376549, PP376550, PP376551, PP376552, PP376553, PP376554, PP376555, PP376556, PP376557, PP376558, PP376559, PP376560, PP376561, PP376562, PP376563, PP376564, PP376565, PP376566, PP376567, PP376568, PP376569, PP376570, PP376571, PP376572, PP376573, PP376574, PP376575, PP376576, PP376577, PP376578, PP376579, PP376580, PP376581, PP376582, PP376583, PP376584, PP376585, PP376586, PP376587, PP376588, PP376589, PP376590, PP376591, PP376592, PP376593, PP376594, PP376595, PP376596, PP376597, PP376598, PP376599, PP376600, PP376601, PP376602, PP376603, PP376604, PP376605, PP376606, PP376607, PP376608, PP376609, PP376610, PP376611, PP376612, PP376613, PP376614, PP376615, PP376616, PP376617, PP376618, PP376619, PP376620, PP376621, PP376622, PP376623, PP376624, PP376625, PP376626, PP376627, PP376628, PP376629, PP376630, PP376631, PP376632, PP376633, PP376634, PP376635, PP376636, PP376637, PP376638, PP376639, PP376640, PP376641, PP376642, PP376643, PP376644, PP376645, PP376646, PP376647, PP376648, PP376649, PP376650, PP376651, PP376652, PP376653, PP376654, PP376655, PP376656, PP376657, PP376658, PP376659, PP376660, PP376661, PP376662, PP376663, PP376664, PP376665, PP376666, PP376667, PP376668, PP376669, PP376670, PP376671, PP376672, PP376673, PP376674, PP376675, PP376676, PP376677, PP376678, PP376679, PP376680, PP376681, PP376682, PP376683, PP376684, PP376685, PP376686, PP376687, PP376688, PP376689, PP376690, PP376691, PP376692, PP376693, PP376694, PP376695, PP376696,

PP376697, PP376698, PP376699, PP376700, PP376701, PP376702, PP376703, PP376704, PP376705, PP376706, PP376707, PP376708, PP376709, PP376710, PP376711, PP376712, PP376713, PP376714, PP376715, PP376716, PP376717, PP376718, PP376719, PP376720, PP376721, PP376722, PP376723, PP376724, PP376725, PP376726, PP376727, PP376728, PP376729, PP376730, PP376731, PP376732, PP376733, PP376734, PP376735, PP376736, PP376737, PP376738, PP376739, PP376740, PP376741, PP376742, PP376743, PP376744, PP376745, PP376746, PP376747, PP376748, PP376749, PP376750, PP376751, PP376752, PP376753, PP376754, PP376755, PP376756, PP376757, PP376758, PP376759, PP376760, PP376761, PP376762, PP376763, PP376764, PP376765, PP376766, PP376767, PP376768, PP376769, PP376770, PP376771, PP376772, PP376773, PP376774, PP376775, PP376776, PP376777, PP376778, PP376779, PP376780, PP376781, PP376782, PP376783, PP376784, PP376785, PP376786, PP376787, PP376788, PP376789, PP376790, PP376791, PP376792, PP376793, PP376794, PP376795, PP376796, PP376797, PP376798, PP376799, PP376800, PP376801, PP376802, PP376803, PP376804, PP376805, PP376806, PP376807, PP376808, PP376809, PP376810, PP376811, PP376812, PP376813, PP376814, PP376815, PP376816, PP376817, PP376818, PP376819, PP376820, PP376821, PP376822, PP376823, PP376824, PP376825, PP376826, PP376827, PP376828, PP376829, PP376830, PP376831, PP376832, PP376833, PP376834, PP376835, PP376836, PP376837, PP376838, PP376839, PP376840, PP376841, PP376842, PP376843, PP376844, PP376845, PP376846, PP376847, PP376848, PP376849, PP376850, PP376851, PP376852, PP376853, PP376854, PP376855, PP376856, PP376857, PP376858, PP376859, PP376860, PP376861, PP376862, PP376863, PP376864, PP376865, PP376866, PP376867, PP376868, PP376869, PP376870, PP376871, PP376872, PP376873, PP376874, PP376875, PP376876, PP376877, PP376878, PP376879, PP376880, PP376881, PP376882, PP376883, PP376884, PP376885, PP376886, PP376887, PP376888, PP376889, PP376890, PP376891, PP376892, PP376893, PP376894, PP376895, PP376896, PP376897, PP376898, PP376899, PP376900, PP376901, PP376902, PP376903, PP376904, PP376905, PP376906, PP376907, PP376908, PP376909, PP376910, PP376911, PP376912, PP376913, PP376914, PP376915, PP376916, PP376917, PP376918, PP376919, PP376920, PP376921, PP376922, PP376923, PP376924, PP376925, PP376926, PP376927, PP376928, PP376929, PP376930, PP376931, PP376932, PP376933, PP376934, PP376935, PP376936, PP376937, PP376938, PP376939, PP376940, PP376941, PP376942, PP376943, PP376944, PP376945, PP376946, PP376947, PP376948, PP376949, PP376950, PP376951, PP376952, PP376953, PP376954, PP376955, PP376956, PP376957, PP376958, PP376959, PP376960, PP376961, PP376962, PP376963, PP376964, PP376965, PP376966, PP376967, PP376968, PP376969, PP376970, PP376971, PP376972, PP376973, PP376974, PP376975, PP376976, PP376977, PP376978, PP376979, PP376980, PP376981, PP376982, PP376983, PP376984, PP376985, PP376986, PP376987, PP376988, PP376989, PP376990, PP376991, PP376992, PP376993, PP376994, PP376995, PP376996, PP376997, PP376998, PP376999, PP377000, PP377001, PP377002, PP377003, PP377004, PP377005, PP377006, PP377007, PP377008, PP377009, PP377010, PP377011, PP377012, PP377013, PP377014, PP377015, PP377016, PP377017, PP377018, PP377019, PP377020, PP377021, PP377022, PP377023, PP377024, PP377025, PP377026, PP377027, PP377028, PP377029, PP377030, PP377031, PP377032, PP377033, PP377034, PP377035, PP377036, PP377037, PP377038, PP377039, PP377040, PP377041, PP377042, PP377043, PP377044, PP377045, PP377046, PP377047, PP377048, PP377049, PP377050, PP377051, PP377052, PP377053, PP377054, PP377055, PP377056, PP377057, PP377058, PP377059, PP377060, PP377061, PP377062, PP377063, PP377064, PP377065, PP377066, PP377067, PP377068, PP377069, PP377070, PP377071, PP377072, PP377073, PP377074, PP377075, PP377076, PP377077, PP377078, PP377079, PP377080, PP377081, PP377082, PP377083, PP377084, PP377085, PP377086,

PP377087, PP377088, PP377089, PP377090, PP377091, PP377092, PP377093, PP377094, PP377095, PP377096, PP377097, PP377098, PP377099, PP377100, PP377101, PP377102, PP377103, PP377104, PP377105, PP377106, PP377107, PP377108, PP377109, PP377110, PP377111, PP377112, PP377113, PP377114, PP377115, PP377116, PP377117, PP377118, PP377119, PP377120, PP377121, PP377122, PP377123, PP377124, PP377125, PP377126, PP377127, PP377128, PP377129, PP377130, PP377131, PP377132, PP377133, PP377134, PP377135, PP377136, PP377137, PP377138, PP377139, PP377140, PP377141, PP377142, PP377143, PP377144, PP377145, PP377146, PP377147, PP377148, PP377149, PP377150, PP377151, PP377152, PP377153, PP377154, PP377155, PP377156, PP377157, PP377158, PP377159, PP377160, PP377161, PP377162, PP377163, PP377164, PP377165, PP377166, PP377167, PP377168, PP377169, PP377170, PP377171, PP377172, PP377173, PP377174, PP377175, PP377176, PP377177, PP377178, PP377179, PP377180, PP377181, PP377182, PP377183, PP377184, PP377185, PP377186, PP377187, PP377188, PP377189, PP377190, PP377191, PP377192, PP377193, PP377194, PP377195, PP377196, PP377197, PP377198, PP377199, PP377200, PP377201, PP377202, PP377203, PP377204, PP377205, PP377206, PP377207, PP377208, PP377209, PP377210, PP377211, PP377212, PP377213, PP377214, PP377215, PP377216, PP377217, PP377218, PP377219, PP377220, PP377221, PP377222, PP377223, PP377224, PP377225, PP377226, PP377227, PP377228, PP377229, PP377230, PP377231, PP377232, PP377233, PP377234, PP377235, PP377236, PP377237, PP377238, PP377239, PP377240, PP377241, PP377242, PP377243, PP377244, PP377245, PP377246, PP377247, PP377248, PP377249, PP377250, PP377251, PP377252, PP377253, PP377254, PP377255, PP377256, PP377257, PP377258, PP377259, PP377260, PP377261, PP377262, PP377263, PP377264, PP377265, PP377266, PP377267, PP377268, PP377269, PP377270, PP377271, PP377272, PP377273, PP377274, PP377275, PP377276, PP377277, PP377278, PP377279, PP377280, PP377281, PP377282, PP377283, PP377284, PP377285, PP377286, PP377287, PP377288, PP377289, PP377290, PP377291, PP377292, PP377293, PP377294, PP377295, PP377296, PP377297, PP377298, PP377299, PP377300, PP377301, PP377302, PP377303, PP377304, PP377305, PP377306, PP377307, PP377308, PP377309, PP377310, PP377311, PP377312, PP377313, PP377314, PP377315, PP377316, PP377317, PP377318, PP377319, PP377320, PP377321, PP377322, PP377323, PP377324, PP377325, PP377326, PP377327, PP377328, PP377329, PP377330, PP377331, PP377332, PP377333, PP377334, PP377335, PP377336, PP377337, PP377338, PP377339, PP377340, PP377341, PP377342, PP377343, PP377344, PP377345, PP377346, PP377347, PP377348, PP377349, PP377350, PP377351, PP377352, PP377353, PP377354, PP377355, PP377356, PP377357, PP377358, PP377359, PP377360, PP377361, PP377362, PP377363, PP377364, PP377365, PP377366, PP377367, PP377368, PP377369, PP377370, PP377371, PP377372, PP377373, PP377374, PP377375, PP377376, PP377377, PP377378, PP377379, PP377380, PP377381, PP377382, PP377383, PP377384, PP377385, PP377386, PP377387, PP377388, PP377389, PP377390, PP377391, PP377392, PP377393, PP377394, PP377395, PP377396, PP377397, PP377398, PP377399, PP377400, PP377401, PP377402, PP377403, PP377404, PP377405, PP377406, PP377407, PP377408, PP377409, PP377410, PP377411, PP377412, PP377413, PP377414, PP377415, PP377416, PP377417, PP377418, PP377419, PP377420, PP377421, PP377422, PP377423, PP377424, PP377425, PP377426, PP377427, PP377428, PP377429, PP377430, PP377431, PP377432, PP377433, PP377434, PP377435, PP377436, PP377437, PP377438, PP377439, PP377440, PP377441, PP377442, PP377443, PP377444, PP377445, PP377446, PP377447, PP377448, PP377449, PP377450, PP377451, PP377452, PP377453, PP377454, PP377455, PP377456, PP377457, PP377458, PP377459, PP377460, PP377461, PP377462, PP377463, PP377464, PP377465, PP377466, PP377467, PP377468, PP377469, PP377470, PP377471, PP377472, PP377473, PP377474, PP377475, PP377476, PP377477, PP377478, PP377479, PP377480, PP377481, PP377482, PP377483, PP377484, PP377485, PP377486, PP377487, PP377488, PP377489, PP377490, PP377491, PP377492, PP377493, PP377494, PP377495, PP377496, PP377497, PP377498, PP377499, PP377500, PP377501, PP377502, PP377503, PP377504, PP377505, PP377506, PP377507, PP377508, PP377509, PP377510, PP377511, PP377512, PP377513, PP377514, PP377515, PP377516, PP377517, PP377518, PP377519, PP377520, PP377521, PP377522, PP377523, PP377524, PP377525, PP377526, PP377527, PP377528, PP377529, PP377530, PP377531, PP377532, PP377533, PP377534, PP377535, PP377536, PP377537, PP377538, PP377539, PP377540, PP377541, PP377542, PP377543, PP377544, PP377545, PP377546, PP377547, PP377548, PP377549, PP377550, PP377551, PP377552, PP377553, PP377554, PP377555, PP377556, PP377557, PP377558, PP377559, PP377560, PP377561, PP377562, PP377563, PP377564, PP377565, PP377566, PP377567, PP377568, PP377569, PP377570, PP377571, PP377572, PP377573, PP377574, PP377575, PP377576, PP377577, PP377578, PP377579, PP377580, PP377581, PP377582, PP377583, PP377584, PP377585, PP377586, PP377587, PP377588, PP377589, PP377590.

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

## Acknowledgements

The authors thank the study participants, their families, and the research staff who contributed data to this analysis. We also thank Sjanna Besteman, Mirjam Hamer, Joanne Wildenbeest, Tessa van Hout, Lies Kriek-Sonius, and Eline Harding for their contribution to the sample collection in the Wilhelmina Children's Hospital, the Netherlands. The INFORM-RSV

study received funding from AstraZeneca and Sanofi. The funders had no role in study design, data collection and analysis, decision to publish, or preparation of the manuscript.

## Author contributions

A.C.L. and L.J.B. conceived the research. A.C.L. and L.J.B. drafted the manuscript with substantial help of B.V. and P.L. B.V. and P.L. performed data analyses along with A.C.L. All authors discussed the results and contributed to the revision of the final manuscript. All authors approved the final version of the manuscript and accept responsibility for the data therein.

## Competing interests

L.J.B. has regular interaction with pharmaceutical and other industrial partners. He has not received personal fees or other personal benefits. U.M.C.U. has received major funding (>€100,000 per industrial partner) for investigator initiated studies from AbbVie, MedImmune, Janssen, the Bill and Melinda Gates Foundation, Nutricia (Danone) and MeMed Diagnostics. U.M.C.U. has received major cash or in kind funding as part of the public private partnership IMI-funded RESCEU project from GSK, Novavax, Janssen, AstraZeneca, Pfizer and Sanofi. U.M.C.U. has received major funding from Julius Clinical for participating in the INFORM-RSV study sponsored by AstraZeneca and Sanofi. U.M.C.U. has received minor funding for participation in trials by Regeneron and Janssen from 2015–2017 (total annual estimate less than €20,000). U.M.C.U. received minor funding for consultation and invited lectures by AbbVie, Med-Immune, Ablynx, Bavaria Nordic, MabXience, Novavax, Pfizer, and Janssen (total annual estimate less than €20,000). L.J.B. is the founding chairman of the ReSViNET Foundation. P.L. and M.A.S. acknowledge support from the European Union's Horizon 2020 research and innovation programme (grant agreement no. 725422-ReservoirDOCS), from the Wellcome Trust through project 206298/Z/17/Z and from the NIH grant R01 AI153044. P.L. acknowledges support from the Research Foundation - Flanders ('Fonds voor Wetenschappelijk Onderzoek - Vlaanderen', G0D5117N and G051322N) and from the European Union's Horizon 2020 project MOOD (grant agreement no. 874850). D.W. and E.J.K. are employees of AstraZeneca. The remaining authors declare no competing interests.

## Additional information

[1]Department of Paediatric Immunology and Infectious Diseases, Wilhelmina Children's Hospital, University Medical Centre Utrecht, Lundlaan 6, 3584 EA Utrecht, the Netherlands. [2]Department of Microbiology, Immunology and Transplantation, Laboratory of Clinical and Epidemiological Virology, Herestraat 49, 3000 Leuven, Belgium. [3]Spatial Epidemiology Lab (SpELL), Université Libre de Bruxelles, Bruxelles, Belgium. [4]Department of Medical Microbiology, University Medical Center Utrecht, Heidelberglaan 100, 3584 CX Utrecht, the Netherlands. [5]Translational Medicine, Vaccines & Immune Therapies, Bio-Pharmaceuticals R&D, AstraZeneca, 1 MedImmune Way, Gaithersburg, MD, USA. [6]Department of Woman's and Child's Health, University Hospital of Padova, Padova, Italy. [7]ReSViNET Foundation, Zeist, the Netherlands. [8]Institute of Pediatric Research "Città della Speranza", Padova, Italy. [9]Jose Eluterio Gonzalez Hospital Universitario, Monterrey, Mexico. [10]Smorodintsev Research Institute of Influenza, St. Petersburg, Russia. [11]Seoul National University Children's Hospital, Seoul, South Korea. [12]Hospital Roberto del Río, Universidad de Chile, Santiago, Chile. [13]MacKay Children's Hospital, New Taipei, Taiwan, ROC. [14]Institute of Virology, University Hospital Giessen and Marburg, Marburg, Germany. [15]Université Paris XII, Créteil, France. [16]McGill University Health Centre, Montreal, QC, Canada. [17]McMaster University, Hamilton, ON, Canada. [18]King's College London, London, UK. [19]University of Western Australia, Perth, WA, Australia. [20]Hospital Clínico Universitario de Santiago, Galicia, Spain. [21]University of Turku and Turku University Hospital, Turku, Finland. [22]Pontificia Universidade Catolica de Rio Grande do Sul, Porto Alegre, Brazil. [23]Fukushima Medical University School of Medicine, Fukushima, Japan. [24]Department of Paediatrics and Child Health, Faculty of Health Sciences, University of the Witwatersrand, Johannesburg, South Africa. [25]South African Medical Research Council, Vaccines & Infectious Diseases Analytics Research Unit, and Department of Science and Technology/National Research Foundation, South African Research Chair Initiative in Vaccine Preventable Diseases, Faculty of Health Sciences, University of the Witwatersrand, Johannesburg, South Africa. [26]Hospices Civils de Lyon and the Centre International de Recherche en Infectiologie (CIRI) Inserm U1111, CNRS UMR5308, ENS de Lyon, UCBL1 Lyon, France. [27]Department of Human Genetics, David Geffen School of Medicine, University of California, Los Angeles, CA 90095, USA. [28]Department of Biostatistics, Jonathan and Karin Fielding School of Public Health, University of California, Los Angeles, CA 90095, USA. [29]Department of Biomathematics, David Geffen School of Medicine, University of California, Los Angeles, CA 90095, USA. [30]Institute for Genomics and Evolutionary Medicine, Department of Biology, Temple University, 801 N Broad St, Philadelphia, PA 19122, USA. [31]INSERM, Sorbonne Université, Institut Pierre Louis d'Epidémiologie et de Santé Publique IPLESP, F75012 Paris, France. [32]These authors contributed equally: Annefleur C. Langedijk, Bram Vrancken. [33]These authors jointly supervised this work: Philippe Lemey, Louis J Bont. ✉e-mail: l.bont@umcutrecht.nl

## the INFORM-RSV Study Group

Annefleur C. Langedijk[1,32], Eugenio Baraldi[6,7,8], Elena Priante[6,8], Abiel Homero Mascareñas de Los Santos[9], Daria M. Danilenko [10], Kseniya Komissarova[10], Eun Hwa Choi[11], Ki Wook Yun[11], María Angélica Palomino[12], Pascale Clement[12], Hsin Chi [13], Christian Keller [14], Monica Bauck[14], Robert Cohen[15], Jesse Papenburg[16], Jeffrey Pernica[17], Anne Greenough [7,18], Atul Gupta[18], Peter Richmond[19], Ushma Wadia[19], Federico Martinón-Torres [7,20], Irene Rivero-Calle[20], Terho Heikkinen[7,21], Renato T. Stein[7,22], Magalia Lumertz[22], Mitsuaki Hosoya[23], Koichi Hasimoto[23], Marta C. Nunes [7,24,25], Charl Verwey[24,26], Shabir A. Madhi[24] & Louis J. Bont [1,7,33]

