## [Peer Review File · Nature Communications]

The Genomic Evolutionary Dynamics and Global Circulation Patterns of Respiratory Syncytial VirusREVIEWER COMMENTS

Reviewer #1 (Remarks to the Author):

The study by Langedijk et al describes the use of RSV sequencing data to study the selective pressures and global movement patterns of RSV. Like for the other respiratory viruses for which these phenomena have been studied, this study finds that RSV evolution is shaped by non-neutral epidemiological processes and that air travel results in substantial global mixing of RSV genotypes. Frankly, anything counter to these findings would have been surprising given the extent to which these drivers have been documented for seasonal influenza and SARS-CoV-2. In fact, given the similarity of the findings presented in this study to those from country-specific studies and to previous large-scale studies that did not rely on whole genome data, I feel like this study is essentially confirmatory in nature, but confirmatory studies do have value and this study appears to be well-executed and technically sound. Given this, most of my comments are about presentation and the substance of the discussion.

1. Lines 110-113: For influenza, local persistence and global circulation were important to resolve because of vaccine strain selection. Why are these important for RSV? The answer is at least partially the near-term arrival of RSV vaccines, but explanation is needed here.
2. Lines 116-117: Why does the genetic sequencing data need to be both prospective and whole genome? Most similar studies for other pathogens have not relied on prospective or whole genome data and the added value is not clear in this study.
3. Line 120: Is this study design genuinely “ideal”? To my mind, ideal would be genuinely global with a lot more virus sequence data over a much longer period of time. Here, the emphasis is on the INFORM-RSV data but ~50% of the data used in this study was collected separately. If this emphasis on INFORM-RSV is to remain, it would seem essential to document how the inclusion of these data inform the results of the study compared to the other ~50% of the data. I.e. how does INFORM-RSV actually change the results? This does eventually appear as a change in statistical support RSVB in figure S3, but is there anything else?
4. Lines 170-172: The authors claim support for a hypothesis described as “our” but have not introduced a hypothesis before this line.
5. Lines 248-257: Here, the authors seem to conflate the INFORM-RSV study and the study described in this manuscript. ~50% of the data used in this study are not part of INFORM-RSV. Unless the INFORM-RSV data alone show support the claims made here, more credit needs to be given to all of the other data creators that made the present study possible.
6. Line 256: Nothing has been “proven”, best case is “shown”. Arguably, other studies have done the same.
7. Lines 259-292: This is the single largest paragraph in this manuscript and it has next-to-nothing to do with the actual findings. It contains unsupported speculation that should be removed. If the authors really want to include this section, they should also include an extensive discussion of feasibility and effectiveness. As demonstrated by numerous studies during the SARS-CoV-2 pandemic and studies on other respiratory viruses before the pandemic, most suggested approaches here are entirely infeasible from a practical perspective and, even if implemented, would have to be enforced in a draconian manner for them to have any effect. I suggest removing this paragraph.
8. Line 295-296: “The finding that RSV spreads by human air travel opens up the possibility for mitigating RSV transmission”. As in comment 7, this statement isn’t true in any practical sense. The importance of air travel for the global spread of seasonal influenza has been understood for ~20 years and well documented for ~10 years and it hasn’t changed anything. This study won’t either. Please remove this sentence.

9. Lines 308-309: "Transform" is an exaggeration. It might be "useful".

10. Lines 328-329: A strength of this study is that it is an improvement on the level of geographic coverage compared to previous studies. A weakness is that it does not cover the vast majority of the world's population or include any of the world's most populous countries.

11. Lines 335-336: Here, the value of expanding surveillance is highlighted because the roll out of maternal and elderly RSV vaccines could increase selective pressure on the virus. It's not impossible for this to be true, but it is highly unlikely. There is no evidence of seasonal influenza virus vaccines shaping influenza virus evolution and these vaccines are widely used. By comparison, the RSV vaccines will be much less widely used and primarily given to populations that are not thought to be the key drivers of virus circulation. The primary value of enhanced RSV surveillance will be periodic updates of the RSV vaccine.

12. Sharing of virus data. Lines 343-344: "the new genome data will constitute a key resource for further extensive research in the field of RSV epidemiology." I hope this statement is true. If it is true, then it is absolutely galling that the data from this study are not publicly available nor are there even accession numbers for where these data may eventually be available. Lines 440-442 suggest that these data might be submitted to genbank when the manuscript is accepted. This was barely acceptable 10 years ago and is wholly unacceptable now, particularly if the authors genuinely believe their data has public health value. The sequencing data from this study should be available before a revision of this manuscript is submitted.

13. Figure 2 purports to show the INFORM-RSV data from 18 sites in 17 countries, and yet many more sites and countries are included. I suspect that this is all of the data included in the study of which ~50% is not INFORM-RSV. Regardless, the description of figure 2 should be amended to reflect its contents.

Reviewer #2 (Remarks to the Author):

Taking advantage of global prospective surveillance of RSV by INFORM-RSV and using whole genome characterization of well-documented cases with respect to location and date of origin, the authors reveal molecular epidemiologic patterns in RSV global evolution consistent with diversifying selection mainly in the G gene, a gene target of potential selection for immune escape. They also show the dominant influence of air travel in RSV molecular epidemiology. This study is powerful and well-executed, with unique strengths that include large sample size, whole genome characterization, and extensive time span of study pre-COVID pandemic (2017-2020). Thus this study contributes importantly and uniquely to our understanding of how RSV evolution is shaped on a global scale, by positive diversifying selection and substantive geographic mixing via air travel, just prior to the availability of monoclonal antibody treatment and vaccines.

Methodologies, statistical and phylogenetic, are very robust and appropriate.

Fig. 1: authors should include node support values for major nodes; x-axis should include a label

Editorial notes:

Line 125: the the

Line 148: clustere

Line 171: use either just "phylogeny" or "phylogenetic trees"

Line 256: replace "proven" with "demonstrated"

Line 448: please report the length of the RSV genome somewhere in this section

Reviewer #3 (Remarks to the Author):

Review of 424141_0

"The Genomic Evolutionary Dynamics and Global Circulation Patterns of Respiratory Syncytial Virus"

This could be a good analysis of important RSV genomic data and may be revealing some features of RSV transmission patterns and evolution. The analysis of 1,282 complete RSV genome sequences from a 3 year period, generated through a common sample collection and sequencing platform would provide a large increase in the sequence data available to examine RSV evolution. However the manuscript is very poorly written, makes a lot of unsupported claims, does not describe the actual analysis in sufficient detail and in many cases presents the data in poorly designed annotated phylogenetic trees that are cluttered and difficult to follow. To tell you the truth, I was initially looking forward to reading the paper because of the importance of the topic and the amount of new RSV data it adds. However I was very disappointed in the quality of the document and I think that the manuscript would benefit from an extensive revision. The text and many of the figures do not do justice to the data. The authors should consider the following points.

1. Figure 2 is pretty but looks like a screen shot of airflight paths from a simple plotting tool. I don't see that any calculated RSV parameters are indicated. The studies included data from 17 countries while Figure 2 shows many more nodes, so apparently these are cities connected to the 17 countries by flights? The Figure 2 legend states "The lines reflect the possible connections of human air travel between INFORM-RSV locations." The actual calculation to generate the map patterns is not described anywhere in the manuscript that I could find. What is actually shown on Figure 2? Again, without explanation this figure is just filler, looks interesting but when examined in more detail many questions arise. It actually is quite misleading as later in the document the authors point to this figure as evidence that it supports their travel conclusions. Line 189: The statement "Our global and temporal collection of RSV sequences provide insights into RSV circulation patterns (Fig.2)" is not supported by the Figure 2. Figure 2 shows only connecting flight paths between the study cities and displays no RSV data so how can it provide insights into RSV circulation patterns?

2. The authors could improve this figure by labelling cities and countries and including important metrics from their study in the map (e.g. coloring the total passenger traffic compared the calculated RSV traffic). Indicating the study cities would help. As it currently stands the figure seems to be empty filler and should be removed.

3. Line 139 " Therefore, we integrated human movement patterns with whole genome sequences from RSV samples that were prospectively collected in 17 countries worldwide over three seasons (2017-2020) prior to the COVID-19 pandemic." "17 countries worldwide over three seasons (2017-2020) "

Define "season". 2017-2020 spans 4 years. RSV infection patterns can differ in North vs South hemisphere so it would be useful if the authors described how these seasons were identified and defined. Also COVID-19 pandemic started at the end of 2019 so some of these collection dates may include the COVID-19 period.

4. The analysis for Figure 3, which is crucial for the conclusions of the paper, that should be better explained to non specialists. For example, an explanation of positive vs negative values for the coefficient * Inclusion value, what does this mean in simple terms. The authors have put very little effort into making their analysis accessible to clinicians or policy makers. If indeed they have discovered some new travel associated pattern of RSV transmission, then an explanation that a policy maker will buy should be given. As currently written, there are probably a dozen phylogeneticists in the world who can follow the analysis and accept the conclusions.

5. Figure 3. The RSV A =, B or Both should be indicated directly on the figure. Color choice makes it difficult to distinguish, the light green and yellow are too similar, the light and dark orange are

too similar.

"geographic distance" is not clear. From center of country or airport? Similar concerns for latitude, what location in each country was used?

6. Figure 3 Legend: "The predictors include air passenger fluxes (air travel, in red)," This should be more clearly explained. Total Number of passengers at both origin and destination? Also from line 532 it simply states "geographic distance" and this is not clear. The authors should specify what location in each country was used to calculate these distances.

7. Figure 3 legend "RSV global diffusion at country level." What is global diffusion at country level? Isn't this country level diffusion?

8. "...applied to both the RSVA and RSVB data sets at the country level" I guess the authors mean applied to the combined RSVA and RSVB data sets?

9. Figure 4, panel A, B should be labeled. I suppose left is RSVA, right is RSVB but this should be spelled out.

Also "The colour and dot size indicate what percentage the mean estimate represents of the maximal mean entropy estimate, ranging from blue (0%) to bright red (100%)." This calculation should be explained in more detail.

10. Figure S4 is misleading. For example. I can see no plausible biological or viral reason why Italy would have fewer unique lineages than France other than Italy reports fewer total RSV genome sequences than France. Also confusing, why would Germany have a larger number of unique RSVA genomes sequences and a very small number of unique RSVB sequences. This is also like to be due to sample bias, Germany data somehow missed peaks of RSVB. An analysis should be run (and reported) plotting number of unique RSVA and RSVB lineages at the start of the most recent season, as a function of total reported sequences for each country. I suspect low number of unique RSVA and RSVB lineages is simply because very few patients were sampled and reported. This bias should be discussed in the manuscript.

11. Line 439. "Whole genome sequencing was performed at the UMC Utrecht using the Illumina NextSeq 500 platform and annotated with sampling data and country. "

Much more detail are needed to explain how the sequences were generated including primers design, the actual sample processing (extraction, RT-PCR, PCR, Illumina library prep, the sequencing itself) the downstream data process from raw Illumina data to assembled genomes, including quality control steps at the raw read level and at the final genome level. None of this is described and should be included in adequate detail to understand how the data were generated.

12. Line 440. "Whole genome sequences derived from the first three seasons of the INFORM-RSV study will be submitted to NCBI GenBank upon publication of this manuscript." These data should have been available at the time of submission of the manuscript for review. The supplemental material should include a table listing for each sequence the date, location and accession number.

13. Line 448 "These were first size-selected (only those of length ≥ 10 k bases were kept for further analyses) and typed as RSV A or RSVB. After alignment with MAFFT v.7.47537 and manual verification using AliView v.1.2638, RDP539 was used to clean the RSV A and RSVB alignments from putative recombinant sequences." The percentage of genomes that passed this quality control should be stated. Also " length ≥ 10 k bases" is not clear. What these should be "length without N ≥ 10 k bases" if that is correct. Otherwise a listing of the number of Ns should also be provided in Table describing the final set of genomes.

14. Line 443 "The resulting alignment served to obtain a maximum likelihood tree with branch support estimated with the SH-aLRT test40 as implemented in IQtree v.2.1.241." The alignment should be made available in supplemental material, or deposited in GitHub or some other public repository.

15. Line 153 "Recently, variants with a duplication in the G

5 gene have emerged 20." This was first reported in 2003, 20 years ago. Perhaps "recently" is not the correct descriptor.

16. The resolution of Figure S1 is horrible and it is not possible to see any of the details of the figure. "The thickness of each branch in the Maximum Clade Credibility (MCC) tree corresponds to the number of AA changes that occurred over that branch." The thickness of most of the branches are "barely legible" so it is difficult to detect changes. The annotations are scattered throughout the figure and it is difficult to know which nodes are being annotated. The phylogenetic tree with this number of nodes simply does not work to present the details of protein changes. The authors should consider presenting the data in a more creative, clearer and more accessible form.

17. Similarly, the resolution of Figure S5 is poor. The big list of location colors is not terrible useful because the colors are so similar. It is not possible to distinguish the country sources based on subtle differences in green shades (Kenya vs Jordan vs Japan...?) or red shades (USA vs UK vs Thailand) Are these distinctions important for the conclusions or just included because the software allowed it? I think the most import cluster is the odd South Africa cluster and I could only see this because of the red circle. So are the other nodes' country origin important? If not, the figure could be simplified and the color palette reduced to red (South Africa) and black (everything else).

18. A listing of the sample collection sites, the city used for geographical distance and latitude, the population values used, number of samples collected, the number of complete genomes (defined by X length, without internal Ns) should be provided in a table.

19. A complete listing of the GenBank accession numbers and the publications for the reference sequences used (not from this study) should be provided in a table.

20. Throughout the "Main" section I keep encountering text that would be more appropriate for Discussion or Introduction sections. It is not really clear what defines this "Main" section. There is in fact no Introduction (which would benefit the article) and it is not clear what decision process was used for including material in the "Main" section. What is wrong with using Intro, Methods, Results, Discussion as described in the Nature Communication guide to authors (<https://www.nature.com/ncomms/submit/article>)? I suggest a revision of the paper to follow the more expected placement of material: background in Introduction, new results in Results, interpretation, caveats etc. in Discussion. e.g. the discussion on human bias due to limited genome numbers on line 199-205 is clearly Discussion material. , lines 108-130, 132-142 are background and should be in Introduction

21. Line 188 The statement "Global RSV circulation patterns are shaped by human air travel" seems to be a statement of fact when it is actually a hypothesis being tested. A question mark should be added to make it clear that this is not a fact.

22. Line 188. What would the RSV evolution patterns look like if global RSV circulation patterns were not shaped by human air travel? If the statement "Global RSV circulation patterns are shaped by human air travel" were posed as a hypothesis, what would the absence of air travel influences look like? I don't disagree that air travel is an important mixer of virus diversity, but I don't see a clear discussion of what how RSV would evolve without human air travel.

23. Is there a non-air travel component to RSV evolution patterns? My concern is that RSV evolves in humans and humans travel extensively by air so of course air travel influences RSV evolution patterns. In the discussion can the authors provide examples of a human virus whose evolution pattern is not influenced by air travel? For example rhinovirus might infect so rapidly and there may be so many local variants and that the introduction by air travel is a trivial component of total rhinovirus evolution.

24. Line 259 "Human air travel increases the likelihood of infectious diseases spreading rapidly between countries and continents." This seems obvious but references should be provided for this statement.

25. Line 260 " It is still unclear how patients acquire viral respiratory disease in the context of air travel" I think there is quite a bit of literature on SARS-CoV-2 acquisition during flights which should be cited (see for example: Transmission of SARS-CoV-2 associated with aircraft travel: a systematic review *J Travel Med.* 2021 Sep 3 :doi: 10.1093/jtm/taab133, Choi EM, Chu D, Cheng P, Tsang D, Peiris M, Bausch DG, et al. In-Flight Transmission of SARS-CoV-2. *Emerg Infect Dis.* 2020;26(11):2713-2716. <https://doi.org/10.3201/eid2611.203254>, Swadi T, Geoghegan JL, Devine T, McElroy C, Sherwood J, Shoemack P, et al. Genomic Evidence of In-Flight Transmission of SARS-CoV-2 Despite Predeparture Testing. *Emerg Infect Dis.* 2021;27(3):687-693. <https://doi.org/10.3201/eid2703.204714> and many more).

26. Line 271 "The best strategy may be to limit the travel of infected passengers by testing at departure or arrival and quarantine infected passengers, as several countries did for COVID-19, and to use airports as sentinel surveillance points as is currently happening in many European countries with testing of Chinese arrivals to monitor variants28."

Are the author proposing testing and travel restrictions for RSV? We already know that this strategy largely failed to limit SARS-CoV-2 spread and it is much better to invest energy into protecting individuals with vaccination. But for a virus that in most individuals is a mild infections, for which a vaccine does not exist, proposing testing and travel restrictions does not seem justified.

27. The entire discussion from line 274 to 292 does not belong in this paper. Nothing in the paper is relevant for travel restrictions or testing or diagnostic methods. And similar to the previous concern, proposing strategies for unworkable and non-functional policies seems out of place in this paper.

28. Line 295 "The finding that RSV spreads by human air travel opens up the possibility for mitigating RSV transmission." Not really, unless societies can do better with mitigating RSV transmission than they did with SARS-CoV-2. Let's be realistic and not make such claims.

Response to comments

Reviewer #1 (Remarks to the Author):

Comment 1 [Main]: Lines 110-113: For influenza, local persistence and global circulation were important to resolve because of vaccine strain selection. Why are these important for RSV? The answer is at least partially the near-term arrival of RSV vaccines, but explanation is needed here.

Response: We agree that the approval of the first-ever RSV vaccines - which may be followed shortly by other vaccines and/or therapeutics - should be mentioned here as this induces a need for monitoring the potential emergence of resistant strains.

Revised text: "With the recent approval of the first-ever RSV vaccines and the monoclonal antibody (mAb) nirsevimab for the prevention of RSV in all infants¹, our understanding of the global transmission dynamics of RSV becomes increasingly important."

Comment 2 [Main]: Lines 116-117: Why does the genetic sequencing data need to be both prospective and whole genome? Most similar studies for other pathogens have not relied on prospective or whole genome data and the added value is not clear in this study.

Response: Reconstructions based on subgenomic fragments (mostly the G gene) do not provide sufficient resolution to accurately track RSV spread at shorter time scales, and may also not adequately capture the evolutionary dynamics. The latter is apparent in RSV genotype classification, where, depending on the used subgenomic fragment, strains can be classified differently, and full genomes provide a more coherent view (Rameakers et al. Virus Evolution 2020) The higher phylogenetic resolution offered by full genomes also improves the evolutionary reconstructions needed to answer fundamental questions such as 'What are the global circulation dynamics of RSV?'. A full genome view is also needed to fully understand the immune evasion repertoire, which may not be restricted to evolution in epitope sites (e.g. Van den Hoecke et al. mBio 2022 for influenza) and can involve sites in different genes. In fact, in this work we found evidence for positive selection acting not only on amino acids in the G gene but also in the L gene (lines 175-180). By referencing to the prospective nature of the INFORM sampling, we aimed to indicate the comprehensive global sampling of circulating RSV lineages. We agree with the Reviewer that this implicit message was not sufficiently clear and have reworded this. All references to 'prospective' have been removed from the text except in the description of the clinical samples (line 461).

Revised text: lines 116 and next: "A challenge for reconstructing viral spread through space and time from genetic data has been the lack of a systematic and comprehensive global sampling of whole genomes from circulating RSV lineages. Current such sampling efforts include the global multiyear multicentre INFORM-RSV study and the Global RSV Surveillance Programme of the World Health Organization (WHO)."

Comment 3 [Main]: Line 120: Is this study design genuinely "ideal"? To my mind, ideal would be genuinely global with a lot more virus sequence data over a much longer period of time. Here, the emphasis is on the INFORM-RSV data but ~50% of the data used in this study was collected separately. If this emphasis on INFORM-RSV is to remain, it would seem essential to document how the inclusion of these data inform the results of the study compared to the other ~50% of the data. I.e. how does INFORM-RSV actually change the results? This does eventually appear as a change in statistical support RSVB in figure S3, but is there anything else?

Response: The INFORM sampling is the first coherent global sampling of RSV full genomes covering multiple successive seasons. The Reviewer correctly notices that the newly

generated data provide the additional power needed to consistently identify air passenger fluxes as a predictor of RSV spread (Fig. S3). In addition to this, we now could also show that the topology of RSVA phylogenies is shaped by non-neutral population turn-over, as opposed to what was previously found (Fig. S1). While these findings may not come across as substantial benefits of the newly obtained data, they represent relevant advances in our understanding of RSV evolution and spread at a global scale. Our findings concerning the added value of the INFORM sampling also reinforce the sometimes undervalued notion that results of statistical inferences can depend on the data at hand. Yet, we fully agree with the Reviewer that referring to the INFORM sampling as 'ideal' is not opportune and removed this notion from the manuscript. We also reworded the first paragraph of the Discussion so as to avoid overly emphasising the role of the newly obtained genomes.

Revised text: lines 120 and next: "The INFORM-RSV study combines large-scale full genome sequencing and a global coverage over multiple RSV seasons to provide a molecular reference of RSV strains and sequence variability."

Discussion, 1st paragraph: "Optimised surveillance and prevention of RSV infection at a global scale relies on our understanding of its spread. Here, we combine existing RSV genomic data and new full genomes from a systematic global sampling effort with empirical data on human mobility, demography and a proxy for synchronicity of RSV seasonality to evaluate which factors shape global RSV circulation. We show that air travel predicts global RSV spread, similar to what has been demonstrated for influenza H3N2^{5,8}, influenza H1N1⁴, and recently SARS-CoV-2⁶. Additional sampling efforts (including those within the framework of the ongoing INFORM-RSV study) are expected to generate more densely sampled genomic data. This will increase the resolution of phylogeographic reconstructions and it will likely allow testing predictors at other spatial scales where other forms of mobility could also shape RSV circulation. Understanding RSV spread is also important in the light of monitoring for escape mutations to emerging prophylactic approaches to RSV, as our findings show these have the potential to spread rapidly on a global scale."

Comment 4 [Comparable site-specific diversifying selection in RSVA and RSVB]: Lines 170-172: The authors claim support for a hypothesis described as "our" but have not introduced a hypothesis before this line.

Response: We agree that this is confusing and have amended the sentence.

Revised text: "Substitutions in positions under positive selection are found on different branches of the phylogeny trees, which is consistent with the expectation of diversifying selection."

Comment 5 [Discussion]: Lines 248-257: Here, the authors seem to conflate the INFORM-RSV study and the study described in this manuscript. ~50% of the data used in this study are not part of INFORM-RSV. Unless the INFORM-RSV data alone show/support the claims made here, more credit needs to be given to all of the other data creators that made the present study possible.

Response: We agree with the reviewer as we have combined INFORM data with historic data. Please see revised text to Comment #3 of this Reviewer.

Comment 6 [Discussion]: Line 256: Nothing has been "proven", best case is "shown". Arguably, other studies have done the same.

Response: This is true and we have toned down our conclusive wording. We have replaced "proven" by "shown".

Revised text: "Understanding RSV spread is also important in the light of monitoring for escape mutations to emerging prophylactic approaches to RSV, as our findings show that

these have the potential to spread rapidly on a global scale.”

Comment 7 [Discussion]: Lines 259-292: This is the single largest paragraph in this manuscript and it has next-to-nothing to do with the actual findings. It contains unsupported speculation that should be removed. If the authors really want to include this section, they should also include an extensive discussion of feasibility and effectiveness. As demonstrated by numerous studies during the SARS-CoV-2 pandemic and studies on other respiratory viruses before the pandemic, most suggested approaches here are entirely infeasible from a practical perspective and, even if implemented, would have to be enforced in a draconian manner for them to have any effect. I suggest removing this paragraph.

Response: This paragraph describes air traffic as a mechanism, but it is true that it is largely speculative. We have revised the paragraph and have reduced the text by $\geq 50\%$.

Revised text: “We speculate that air traffic could be a mechanism of RSV transmission.”

Comment 8 [Discussion]: Line 295-296: “The finding that RSV spreads by human air travel opens up the possibility for mitigating RSV transmission”. As in comment 7, this statement isn’t true in any practical sense. The importance of air travel for the global spread of seasonal influenza has been understood for ~20 years and well documented for ~10 years and it hasn’t changed anything. This study won’t either. Please remove this sentence.

Response: We have removed the sentence per the suggestion of the reviewer.

Comment 9 [Discussion]: Lines 308-309: “Transform” is an exaggeration. It might be “useful”.

Response: We have toned down the sentence.

Revised text: “Therefore, genotyping based on complete genome sequences, instead of genotyping based on nucleotide sequence variability of subgenomic regions (mostly the G gene), can improve the RSV surveillance field by providing a more coherent classification.”

Comment 10 [Discussion]: Lines 328-329: A strength of this study is that it is an improvement on the level of geographic coverage compared to previous studies. A weakness is that it does not cover the vast majority of the world’s population or include any of the world’s most populous countries.

Response: We agree with the Reviewer that should also highlight the broad geographic coverage as a strength of this work. We have revised the manuscript.

Revised text: (lines 329-330): “Strengths of this study are the sample size, the use of complete genomes, and a wide geographic coverage over a period of many years.”

Comment 11 [Discussion]: Lines 335-336: Here, the value of expanding surveillance is highlighted because the roll out of maternal and elderly RSV vaccines could increase selective pressure on the virus. It’s not impossible for this to be true, but it is highly unlikely. There is no evidence of seasonal influenza virus vaccines shaping influenza virus evolution and these vaccines are widely used. By comparison, the RSV vaccines will be much less widely used and primarily given to populations that are not thought to be the key drivers of virus circulation. The primary value of enhanced RSV surveillance will be periodic updates of the RSV vaccine.

Response: We understand this comment. However, the world is about to introduce a monoclonal antibody against a virus to all children born for the first time. Escape from an antibody directed against a single epitope is much more likely than from a regular vaccine. We have explained this in the revised manuscript.

Revised text: “Surveillance of RSV may be particularly important in the wake of recently

approved mAbs and vaccines, though less likely, given the potential for increased immunologic pressure on RSV F.”

Comment 12 [Discussion]: Sharing of virus data. Lines 343-344: “the new genome data will constitute a key resource for further extensive research in the field of RSV epidemiology.” I hope this statement is true. If it is true, then it is absolutely galling that the data from this study are not publicly available nor are there even accession numbers for where these data may eventually be available. Lines 440-442 suggest that these data might be submitted to genbank when the manuscript is accepted. This was barely acceptable 10 years ago and is wholly unacceptable now, particularly if the authors genuinely believe their data has public health value. The sequencing data from this study should be available before a revision of this manuscript is submitted.

Response: We have added the list of GISAID accession numbers to the manuscript. Next to this, the alignments and BEAST XML files used for the phylogeographic and selection analyses are now also publicly available via a GitHub repository (https://github.com/bramvrancken/RSV_INFORM).

Comment 13 [Figures]: Figure 2 purports to show the INFORM-RSV data from 18 sites in 17 countries, and yet many more sites and countries are included. I suspect that this is all of the data included in the study of which ~50% is not INFORM-RSV. Regardless, the description of figure 2 should be amended to reflect its contents.

Response: We thank the Reviewer for pointing this out. As this Figure indeed contributes little, we decided to remove it from the paper, and removed all references to this Figure in the revised version.

Reviewer #2 (Remarks to the Author):

Comment 1 [Figure 1]: Fig. 1: authors should include node support values for major nodes; x-axis should include a label.

Response: We included the axis label in the Figure. Concerning the genotype clade support, which for all genotypes is a combination of SH-aLRT and UFB support values of >80 and >90 respectively (see also lines 468-469, Methods), we opted to include these in a Supplementary Table so as to avoid adding further information to what are already rather dense visuals. A reference to this Supplementary Table has been included in the caption of Figure 1.

Comment 2 [General]: Editorial notes:

Line 125: the the

Line 148: clustere

Line 171: use either just “phylogeny” or “phylogenetic trees”

Line 256: replace “proven” with “demonstrated”

Line 448: please report the length of the RSV genome somewhere in this section.

Response: The text has been updated according to the Reviewer's suggestions. The length of the RSV genome is now mentioned in the second sentence of this paragraph.

Reviewer #3 (Remarks to the Author):

General comment: This could be an good analysis of important RSV genomic data and may be revealing some features of RSV transmission patterns and evolution. The analysis of 1,282

complete RSV genome sequences from a 3 year period, generated through a common sample collection and sequencing platform would provides a large increase in the sequence data available to examine RSV evolution. However the manuscript is very poorly written, makes a lot of unsupported claims, does not describe the actual analysis in sufficient detail and in many cases presents the data in poorly designed annotated phylogenetic trees that are cluttered and difficult to follow. To tell you the truth, I was initial looking forward to reading the paper because of the importance of the topic and the amount of of new RSV data it adds. However I was very disappointed in the quality of the document and I think that the manuscript would benefit from an extensive revision. The text and many of the figures do not do justice to the data. The authors should consider the following points.

Response: We regret that the reviewer has raised important points of critique which we taken to heart. We are specifically grateful to all valuable suggestions the reviewer made. In our revision we have improved the structure and figures of the manuscript. We believe that these changes have improved the quality. We hope the reviewer agrees with the suggested changes.

Comment 1 [Figure 2]: Figure 2 is pretty but looks like a screen shot of airflight paths from a simple plotting tool. I don't see that any calculated RSV parameters are indicated. The studies included data from 17 countries while Figure 2 shows many more nodes, so apparently these are cities connected to the 17 countries by flights? The Figure 2 legend states "The lines reflect the possible connections of human air travel between INFORM-RSV locations." The actual calculation to generate the map patterns is not described anywhere in the manuscript that I could find. What is actually shown on Figure 2? Again, without explanation this figure is just filler, looks interesting but when examined in more detail many questions arise. It actually is quite misleading as later in the document the authors point to this figure as evidence that it supports their travel conclusions. Line 189: The statement "Our global and temporal collection of RSV sequences provide insights into RSV circulation patterns (Fig.2)" is not supported by the Figure 2. Figure 2 shows only connecting flight paths between the study cities and displays no RSV data so how can it provide insights into RSV circulation patterns?

Response: We agree with the reviewer that figure 2 does not add sufficiently to the paper. We also refer to our answer to Comment #13 of Reviewer 1. We have removed the figure.

Comment 2 [Figure 2]: The authors could improve this figure by labelling cities and countries and including important metrics from their study in the map (e.g. coloring the total passenger traffic compared the calculated RSV traffic). Indicating the study cities would help. As it currently stands the figure seems to be empty filler and should be removed.

Response: We agree with the reviewer and refer to our answer to Comment #13 of Reviewer 1. We have removed the figure.

Comment 3 [Main]: Line 139: " Therefore, we integrated human movement patterns with whole genome sequences from RSV samples that were prospectively collected in 17 countries worldwide over three seasons (2017-2020) prior to the COVID-19 pandemic." "17 countries worldwide over three seasons (2017-2020) " Define "season". 2017-2020 spans 4 years. RSV infection patterns can differ in North vs South hemisphere so it would be useful if the authors described how these seasons were identified and defined. Also COVID-19 pandemic started at the end of 2019 so some of these collection dates may include the COVID-19 period.

Response: We agree this was not sufficiently clear in the original manuscript. COVID-19 only led to severe interruptions of (global) mobility as of January 2020. The overlap of travel restrictions will only affect the most recent samples, and hence cannot have interfered substantially with the GLM analyses. We have clarified this in the revised manuscript.

Revised text: “Therefore, we integrated human movement patterns with whole genome sequences from RSV samples that were collected in 17 countries worldwide over three RSV seasons (2017-2020) prior to the COVID-19 pandemic. Travel restrictions due to COVID-19 have not affected the current analysis.”

Comment 4 [Figure 3]: The analysis for Figure 3, which is crucial for the conclusions of the paper, that should be better explained to non specialists. For example, an explanation of positive vs negative values for the coefficient * Inclusion value, what does this mean in sim[le] terms. The authors have put very little effort into making their analysis accessible to clinicians or policy makers. If indeed they have discovered some new travel associate pattern of RSV transmission, then an explanation that a policy make will buy should be given. As currently written, there are probably a dozen phylogeneticists in the world who can follow the analysis and accept the conclusions.

Response: We agree with the lack of explanation for the non-specialists. Although the phylogeographic approach, including the GLM parameterization for predictor identification, is relatively widely used in the last decade, we understand that its introduction to a new field should be accommodated with better explanation. In addition to the detailed explanation in the Methods section, we now also explain the general approach in lay terms when we present the results. This includes text on how to interpret the estimates.

Revised text: “To explore the factors that shape RSV global circulation, we apply a Bayesian phylogeographic approach that models the movement of virus lineages between a set of discrete locations²⁰. This process is generally parameterized in terms of transition rates for all pairs of locations. Here, we use an extension of the discrete phylogeographic model that parameterizes these transition rates as a function of a number of potential predictors⁵. This generalized linear model (GLM) parameterization allows estimating the contribution of each predictor to the spatial diffusion as a coefficient (on a log scale). In addition, the model includes boolean indicator variables that determine the in- or exclusion of predictors allowing to estimate their inclusion probability. Here, we report the posterior distribution of the product of the log coefficient and inclusion probability for each predictor; positive estimates indicate a positive association between predictors and diffusion intensity while the opposite is true for negative estimates. As predictors, we consider human air travel, population size, geographic distances, and latitude differences (see Methods). Our analyses consistently support human air travel as a strong predictor of RSV global spread at both the country (strongly positive estimates, Fig.2) and continental level (Fig. S4) for RSVA and RSVB separately, as well as for a model applied to both RSVA and RSVB data sets combined.”

Comment 5 [Figure 3]: Figure 3. The RSV A, B or Both should be indicated directly on the figure. Color choice makes it difficult to distinguish, the light green and yellow are too similar, the light and dark orange are too similar.

“geographic distance” is not clear. From center of country or airport? Similar concerns for latitude, what location in each country was used?

Response: All figures reporting results of the GLM analyses have been adjusted in line with the suggestions by the Reviewer. We now detail this in the Methods section.

Revised text: “For the geographic distances and absolute latitude differences, latitude and longitude coordinates representing the countries' midpoints taken from the Dataset Publishing Language as provided by Google⁵³ were used. Geographic distances were calculated using the Haversine formula.”

Comment 6 [Figure 3]: Figure 3 Legend: “The predictors include air passenger fluxes (air travel, in red),” This should be more clearly explained. Total Number of passengers at both origin and destination? Also from line 532 it simply states “geographic distance” and this is

not clear. The authors should specify what location in each country was used to calculate these distances.

Response: We now detail in the Methods section that air passenger fluxes refer to the number of passengers traveling between each pair of countries and continents, and also detail this in the caption of Figures 2, S4 and S5. We refer to our reply to Comment #5 of this Reviewer for the added detail concerning how geographic distances and latitude differences were obtained.

Revised text: Methods: “As predictors, we included passenger fluxes (i.e. the number of passengers travelling by air between countries and continents provided by the International Air Transport Association (IATA)⁵⁵ for the period 2019-2020), ...

Figure captions: The predictors include the number of passengers travelling by air between each pair of countries represented in the data set (air travel, in dark red), ...”

Comment 7 [Figure 3]: Figure 3 legend “RSV global diffusion at country level.” What is global diffusion at country level? Isn’t this country level diffusion?

Response: The caption was updated according to the Reviewers' suggestion.

Revised text: “The relevant sentences now read: “Posterior estimates of time-homogeneous predictor contributions to RSV diffusion between countries.”

Comment 8 [Figure S3]: “...applied to both the RSVA and RSVB data sets at the country level” I guess the authors mean applied to the combined RSVA and RSVB data sets?

Response: The text in the captions of this Figure and Figure S5 was updated according to the Reviewer's suggestion.

Revised text: “The concerning sentences now read: “E and F summarize the estimates for a single GLM-diffusion model applied to the combined RSVA and RSVB data sets at the country level.”

Comment 9 [Figure 4]: Figure 4, panel A, B should be labeled. I suppose left is RSVA, right is RSVB but this should be spelled out.

Also “The colour and dot size indicate what percentage the mean estimate represents of the maximal mean entropy estimate, ranging from blue (0%) to bright red (100%).” This calculation should be explained in more detail.

Response: We agree with the reviewer that labelling the panels in this figure required revision. We now also explain in more detail how the size and color of the circles indicating the mean normalised entropy values were derived.

Revised text: “The size of the circles is proportional to what fraction of the highest mean estimate each average estimate represents. The same is indicated by the colors of the circles, which range from blue for an average estimate that represents 0% of the highest value to bright red for the highest mean estimate.”

Comment 10 [Figure S4]: Figure S4 is misleading. For example. I can see no plausible biological or viral reason why Italy would have fewer unique lineages than France other than Italy reports fewer total RSV genome sequences than France. Also confusing, why would Germany have a larger number of unique RSVA genomes sequences and a very small number of unique RSVB sequences. This is also like to be due to sample bias, Germany data somehow missed peaks of RSVB. An analysis should be run (and reported) plotting number of unique RSVA and RSVB lineages at the start of the most recent season, as a function of total reported sequences for each country. I suspect low number of unique RSVA and RSVB lineages is simply because very few patients were sampled and reported. This bias should be discussed in the manuscript.

Response: We agree with the Reviewer that the sampling process can impact the

order/ranking of the number of unique lineages by country. As Figure S4 contributes only little to the interpretation of results, we opted to remove it from the manuscript. Instead, we now provide the location-annotated MCC summary trees from the RSVA and RSVB analyses on which the figure was based, and which can be investigated with FigTree (<http://tree.bio.ed.ac.uk/software/figtree/>), as supplementary Files, and refer to these in the Results section.

Revised text: “Specifically, we summarise normalised entropy measures or the phylogeographic clustering by country, reflecting the degree of phylogenetic interspersion of country-specific lineages (Fig.4), and the number of unique lineages associated with each country circulating at the start of the most recent RSV season (see Supplementary Files S1 and S2 for the MCC summary trees from the evolutionary reconstructions underlying these inferences).”

Comment 11 [Methods]: Line 439. “Whole genome sequencing was performed at the UMC Utrecht using the Illumina NextSeq 500 platform and annotated with sampling data and country. “Much more detail are needed to explain how the sequences were generated including primers design, the actual sample processing (extraction, RT-PCR, PCR, Illumina library prep, the sequencing itself) the downstream data process from raw Illumina data to assembled genomes, including quality control steps at the raw read level and at the final genome level. None of this is described and should be included in adequate detail to understand how the data were generated.

Response: We agree with the Reviewer that such details are essential. We have now emphasized that these details have been published in a separate methodology paper (Langedijk et al BMC Infect Dis 2020).

Revised text: “Whole genome sequencing was performed at the UMC Utrecht using the Illumina NextSeq 500 platform (details have been published a separate methodology paper²) and annotated with sampling data and country.”

Comment 12 [Methods]: Line 440. “Whole genome sequences derived from the first three seasons of the INFORM-RSV study will be submitted to NCBI GenBank upon publication of this manuscript.” These data should have been available at the time of submission of the manuscript for review. The supplemental material should include a table listing for each sequence the date, location and accession number.

Response: We refer to our answer to Comment #13 of Reviewer 1.

Comment 13 [Methods]: Line 448 “These were first size-selected (only those of length \geq 10k bases were kept for further analyses) and typed as RSV A or RSVB. After alignment with MAFFT v.7.47537 and manual verification using AliView v.1.2638, RDP539 was used to clean the RSV A and RSVB alignments from putative recombinant sequences.” The percentage of genomes that passed this quality control should be stated. Also “length \geq 10k bases” is not clear. What these should be “length without N \geq 10k bases” if that is correct. Otherwise a listing of the number of Ns should also be provided in Table describing the final set of genomes.

Response: We indeed filtered sequences with less than 10k non-N characters, and now provide information about the number of sequences that were removed by this filtering step.

Revised text: “These were first size-selected (only those of length without N \geq 10k bases were kept for further analyses, $n = 2865/27417$ or 10.4%) and typed as RSVA or RSVB.”

Comment 14 [Methods]: Line 443 “The resulting alignment served to obtain a maximum likelihood tree with branch support estimated with the SH-aLRT test⁴⁰ as implemented in

IQtree v.2.1.241.” The alignment should be made available in supplemental material, or deposited in GitHub or some other public repository.

Response: We agree with the Reviewer and provide the alignments in a public GitHub repository, to which we refer in a data availability paragraph. We now also include Figures (S7 and S8) indicating the well-supported RSV A and B clades containing all INFORM sequences.

Revised text: “From this tree, a well-supported subtree containing all INFORM-RSV sequences was selected for downstream analyses (Figures S7 and S8).”

Comment 15 [Main]: Line 153 “Recently, variants with a duplication in the G 5 gene have emerged 20.” This was first reported in 2003, 20 years ago. Perhaps “recently” is not the correct descriptor.

Response: We agree and removed the word “recently”.

Comment 16 [Figure S1]: The resolution of Figure S1 is horrible and it is not possible to see any of the details of the figure. “The thickness of each branch in the Maximum Clade Credibility (MCC) tree corresponds to the number of AA changes that occurred over that branch.” The thickness of most of the branches are “barely legible” so it is difficult to detect changes. The annotations are scattered throughout the figure and it is difficult to know which nodes are being annotated. The phylogenetic tree with this number of nodes simply does not work to present the details of protein changes. The authors should consider presenting the data in a more creative, clearer and more accessible form.

Response: To improve the legibility of the annotated trees in this Figure, we opted to provide the RSVA and RSVB trees as separate Figures, and changed the format from PDF to TIFF. To make the link between the branch thickness and the number of changes clearer, we also updated the legend such that the thickness of the lines corresponds to the number of changes accommodated by the branch, and point this out in the figure caption. Furthermore, the PDF version of these figures - which allow zooming in at the desired level of detail - is now also provided in a public GitHub repository.

Revised text: (Figures S1 and S2, caption): “The thickness of each branch in the Maximum Clade Credibility (MCC) tree corresponds to the number of AA changes that occurred over that branch as indicated in the legend.”

Comment 17 [Figure S5]: Similarly, the resolution of Figure S5 is poor. The big list of location colors is not terribly useful because the colors are so similar. It is not possible to distinguish the country sources based on subtle differences in green shades (Kenya vs Jordan vs Japan...?) or red shades (USA vs UK vs Thailand) Are these distinctions important for the conclusions or just included because the software allowed it? I think the most important cluster is the odd South Africa cluster and I could only see this because of the red circle. So are the other nodes’ country origin important? If not, the figure could be simplified and the color palette reduced to red (South Africa) and black (everything else).

Response: We agree with the Reviewer and now present the tree with a simplified color scheme as suggested.

Comment 18 [General]: A listing of the sample collection sites, the city used for geographical distance and latitude, the population values used, number of samples collected, the number of complete genomes (defined by X length, without internal Ns) should be provided in a table.

Response: We now provide information on the sampling locations, the number of sequences by country and their length as tab-delimited files in a public GitHub repository. We now also publicly share the alignments, location-specific predictor data and the BEAST XML files used

for the GLM-analyses, which include all the predictor data (such as the air transportation data). The alignments and BEAST XML files used for the selection analyses are also shared. We refer to this public GitHub repository in a Data Availability paragraph in the Methods section. We also refer to our reply to Comment 5 from this Reviewer for information on how the geographical and latitude distances were obtained.

Revised text: “Alignments, predictor data and BEAST XML files used for this work are available on GitHub (https://github.com/bramvrancken/RSV_INFORM).”

Comment 19 [General]: A complete listing of the GenBank accession numbers and the publications for the reference sequences used (not from this study) should be provided in a table.

Response: The publicly available sequences that were used for this study are available from the alignments provided in the newly created GitHub repository (see our reply to the previous comment of this Reviewer). All sequences can uniquely be identified by their GISAID accession number, which is consistently the first element in the sequence headers.

Comment 20 [Main]: Throughout the “Main” section I keep encountering text that would be more appropriate for Discussion or Introduction sections. It is not really clear what defines this “Main” section. There is in fact no Introduction (which would benefit the article) and it is not clear what decision process was used for including material in the “Main” section. What is wrong with using Intro, Methods, Results, Discussion as described in the NatureCommunication guide to authors (<https://www.nature.com/ncomms/submit/article>)?

I suggest a revision of the paper to follow the more expected placement of material: background in Introduction, new results in Results, interpretation, caveats etc. in Discussion. e.g. the discussion on human bias due to limited genome numbers on line 199-205 is clearly Discussion material. , lines 108-130, 132-142 are background and should be in Introduction.
Response: Originally we submitted our manuscript to Nature after which it was automatically transferred to Nature Communications. We have now followed the Nature Communications guidelines.

Comment 21 [Global RSV circulation patterns are shaped by human air travel]: Line 188 The statement “Global RSV circulation patterns are shaped by human air travel” seems to be a statement of fact when it is actually a hypothesis being tested. A question mark should be added to make it clear that this is not a fact.

Response: We have followed the guidelines of Nature Communications to use subheadings in the Results section (<https://www.nature.com/documents/ncomms-submission-guide.pdf>). These subheadings also frequently summarize the findings of a subsection or highlight the main finding in the subsection. Therefore, we opted to keep these subheadings.

Comment 22 [Global RSV circulation patterns are shaped by human air travel]: Line 188.

What would the RSV evolution patterns look like if global RSV circulation patterns were not shaped by human air travel? If the statement “Global RSV circulation patterns are shaped by human air travel” were posed as a hypothesis, what would the absence of air travel influences look like? I don’t disagree that air travel is an important mixer of virus diversity, but I don’t see a clear discussion of what how RSV would evolve without human air travel.

Response: We agree with the reviewer and have added on this in the Discussion. Please see the revised text to comment 23 below.

Comment 23 [Discussion]: Is there a non-air travel component to RSV evolution patterns? My concern is that RSV evolves in humans and humans travel extensively by air so of course air

travel influences RSV evolution patterns. In the discussion can the authors provide examples of a human virus whose evolution pattern is not influenced by air travel? For example rhinovirus might infect so rapidly and there may be so many local variants and that the introduction by air travel is a trivial component of total rhinovirus evolution.

Response: With this data set we find that none of the other tested predictors consistently contributes significantly to the spread of RSV at a global scale. We expect, however, that additional sampling (including the ongoing efforts within the framework of the INFORM project) will result in more densely sampled genomes by location. This will increase the resolution of phylogeographic reconstructions and it will likely allow testing predictors at other spatial scales, e.g. within continents for which other forms of mobility may also contribute to RSV circulation. For this reason, we consider this work as a first step towards more fine-grained analyses of RSV spread. We have now added a note on this to the Discussion.

Revised text: (first paragraph Discussion): “Additional sampling efforts (including those within the framework of the ongoing INFORM project) are expected to generate more densely sampled genomic data. This will increase the resolution of phylogeographic reconstructions and it will likely allow testing predictors at other spatial scales where other forms of mobility could also shape RSV circulation.”

Comment 24 [Discussion]: Line 259 “Human air travel increases the likelihood of infectious diseases spreading rapidly between countries and continents.” This seems obvious but references should be provided for this statement.

Response: We have now referenced this statement.

Comment 25 [Discussion]: Line 260 “ It is still unclear how patients acquire viral respiratory disease in the context of air travel” I think there is quite a bit of literature on SARS-CoV-2 acquisition during flights which should be cited (see for example: Transmission of SARS-CoV-2 associated with aircraft travel: a systematic review J Travel Med. 2021 Sep 3 :doi: 10.1093/jtm/taab133, Choi EM, Chu D, Cheng P, Tsang D, Peiris M, Bausch DG, et al. In-Flight Transmission of SARS-CoV-2. Emerg Infect Dis. 2020;26(11):2713-2716. <https://doi.org/10.3201/eid2611.203254>, Swadi T, Geoghegan JL, Devine T, McElroy C, Sherwood J, Shoemack P, et al. Genomic Evidence of In-Flight Transmission of SARS-CoV-2 Despite Predeparture Testing. Emerg Infect Dis. 2021;27(3):687-693. <https://doi.org/10.3201/eid2703.204714> and many more).

Response: We thank the reviewer for providing these articles and have implemented them in the discussion section.

Revised text: “Other evidence suggests that SARS-CoV-2 is transmitted during air travel^{24,25}.”

Comment 26 [Discussion]: Line 271 “The best strategy may be to limit the travel of infected passengers by testing at departure or arrival and quarantine infected passengers, as several countries did for COVID-19, and to use airports as sentinel surveillance points as is currently happening in many European countries with testing of Chinese arrivals to monitor variants²⁸.”

Are the author proposing testing and travel restrictions for RSV? We already know that this strategy largely failed to limit SARS-CoV-2 spread and it is much better to invest energy into protecting individuals with vaccination. But for a virus that in most individuals is a mild infections, for which a vaccine does not exist, proposing testing and travel restrictions does not seem justified.

Response: We have removed this paragraph based on this comment and comment 3 of Reviewer 1.

Comment 27 [Discussion]: The entire discussion from line 274 to 292 does not belong in this paper. Nothing in the paper is relevant for travel restrictions or testing or diagnostic methods. And similar to the previous concern, proposing strategies for unworkable and non-functional policies seems out of place in this paper.

Response: We have removed this paragraph as suggested.

Comment 28 [Discussion]: Line 295 “The finding that RSV spreads by human air travel opens up the possibility for mitigating RSV transmission.” Not really, unless societies can do better with mitigating RSV transmission than they did with SARS-CoV-2. Let’s be realistic and not make such claims.

Response: We have removed this statement from the manuscript.

Additional updates:

1. During the revision we noticed that a wrong reference was made to Figure S1 in the submitted manuscript. We have included the correct figure (Figure S3 in the revised manuscript).

REVIEWERS' COMMENTS

Reviewer #1 (Remarks to the Author):

I appreciate the authors' careful consideration of my feedback. The revised manuscript addresses all of my concerns.

Reviewer #3 (Remarks to the Author):

The work has been extensively revised and the authors have seriously considered my concerns and addressed them adequately.